# Cooling positronium to ultralow velocities with a chirped laser pulse train

K. Shu[1,2], Y. Tajima[2], R. Uozumi[2], N. Miyamoto[2], S. Shiraishi[2], T. Kobayashi[2], A. Ishida[3✉], K. Yamada[3], R. W. Gladen[3], T. Namba[4], S. Asai[3], K. Wada[5], I. Mochizuki[5], T. Hyodo[5], K. Ito[6], K. Michishio[6], B. E. O'Rourke[6], N. Oshima[6] & K. Yoshioka[1,2✉]

When laser radiation is skilfully applied, atoms and molecules can be cooled[1–3], allowing the precise measurements and control of quantum systems. This is essential for the fundamental studies of physics as well as practical applications such as precision spectroscopy[4–7], ultracold gases with quantum statistical properties[8–10] and quantum computing. In laser cooling, atoms are slowed to otherwise unattainable velocities through repeated cycles of laser photon absorption and spontaneous emission in random directions. Simple systems can serve as rigorous testing grounds for fundamental physics—one such case is the purely leptonic positronium[11,12], an exotic atom comprising an electron and its antiparticle, the positron. Laser cooling of positronium, however, has hitherto remained unrealized. Here we demonstrate the one-dimensional laser cooling of positronium. An innovative laser system emitting a train of broadband pulses with successively increasing central frequencies was used to overcome major challenges posed by the short positronium lifetime and the effects of Doppler broadening and recoil. One-dimensional chirp cooling was used to cool a portion of the dilute positronium gas to a velocity distribution of approximately 1 K in 100 ns. A major advancement in the field of low-temperature fundamental physics of antimatter, this study on a purely leptonic system complements work on antihydrogen[13], a hadron-containing exotic atom. The successful application of laser cooling to positronium affords unique opportunities to rigorously test bound-state quantum electrodynamics and to potentially realize Bose–Einstein condensation[14–18] in this matter–antimatter system.

The cooling of positronium (Ps) has profound implications for fundamental physics. As the simplest atomic system, consisting of a bound state of two leptons, Ps can serve as a rigorous testing ground for quantum electrodynamics[19], one of the most precisely verified theories of modern physics. An example is the measurement of the $1S$–$2S$ transition frequency[20–22], in which a reduction in the present fractional uncertainty of $2.6 \times 10^{-9}$ is required for a stringent comparison with quantum electrodynamics calculations accurate to $4.7 \times 10^{-10}$ (refs. 23–25). Reducing systematic errors using cold Ps gas and an optical frequency comb is essential. Moreover, cold Ps atoms can provide a unique experimental platform to search for charge–parity–time reversal (CPT) symmetry breaking in the lepton sector, and investigating the effects of gravity on antimatter[26–33]. Owing to its light mass (twice that of an electron), Bose–Einstein condensation (BEC) is expected to occur at relatively high temperatures, ranging from a few kelvin to several tens of kelvin[14–18], compared with ordinary atoms. Discussions regarding the generation of coherent γ-rays using Ps BECs have been noted[34,35]. Furthermore, cold Ps can be used for efficient antihydrogen generation[36]. The cooling of Ps is a requisite for such investigations.

Despite these high expectations, there has been no demonstration of laser cooling of Ps. The concept of laser cooling of Ps using the $1S$–$2P$ transition with a natural linewidth of 50 MHz was first explored some 30 years ago[37], with some preliminary implementation studies[38,39] thereafter. Laser cooling with a wide spectral width and long laser duration in a high magnetic field has recently been investigated theoretically[40].

Laser cooling is an established technique applied to atoms and molecules, based on photon recoil and the Doppler effect. The fundamental mechanism is the reduction in the translational momentum of a moving particle resulting from the absorption of a red-detuned laser photon that propagates counter to the motion and the subsequent emission of a photon in a random direction. Cooling is achieved through repeated cycles of this absorption–emission process. Examples of current laser cooling techniques include Doppler cooling, chirp cooling and magneto-optical traps.

Two key characteristics of Ps make its laser cooling challenging—its brief lifetime and the large recoil-induced frequency shift. The triplet $1S$ state of Ps has a lifetime of 142 ns due to pair annihilation into γ-rays, thereby requiring rapid cooling. Furthermore, as Ps is nearly three

[1]Photon Science Centre, School of Engineering, The University of Tokyo, Yayoi, Bunkyo-ku, Tokyo, Japan. [2]Department of Applied Physics, School of Engineering, The University of Tokyo, Hongo, Bunkyo-ku, Tokyo, Japan. [3]Department of Physics, Graduate School of Science, The University of Tokyo, Hongo, Bunkyo-ku, Tokyo, Japan. [4]International Centre for Elementary Particle Physics (ICEPP), The University of Tokyo, Hongo, Bunkyo-ku, Tokyo, Japan. [5]Institute of Materials Structure Science, High Energy Accelerator Research Organization (KEK), Tsukuba, Ibaraki, Japan. [6]National Institute of Advanced Industrial Science and Technology (AIST), Umezono, Tsukuba, Ibaraki, Japan. ✉e-mail: ishida@icepp.s.u-tokyo.ac.jp; yoshioka@fs.t.u-tokyo.ac.jp

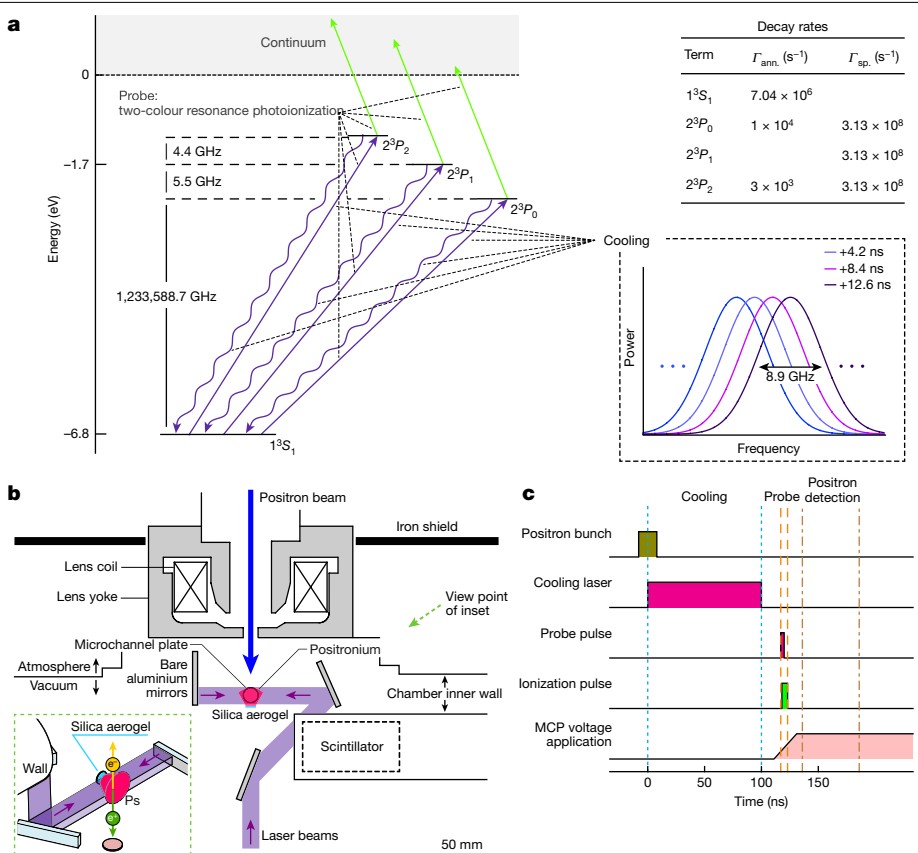

**Fig. 1 | Experimental setup. a**, Interactions between the laser pulses and Ps. Relevant energy levels are shown up to the total angular momentum. The cooling pulses simultaneously address all the 2*P* sublevels, followed by relaxation to the 1*S* level by spontaneous emission. In the Doppler profile measurements, we simultaneously excite all the 2*P* sublevels with the probe pulse. Following excitation, an ionization laser pulse at 532 nm produces photoionized electrons and positrons from the excited Ps atoms in the 2*P* state. The frequencies shown in the figure correspond to the difference in eigenenergies[23]. The table shows the decay rate for each state. $\Gamma_{\text{ann.}}$ and $\Gamma_{\text{sp.}}$ are the decay rates for the annihilation and spontaneous emission, respectively. Inset in dashed lines, the spectrum of the constituent pulses of the cooling laser changes with time. **b**, Top view of the experimental setup in the vacuum chamber. The inset shows a bird's eye view of the setup. **c**, Timing chart illustrating the sequence of positron injection, laser cooling of Ps and detection of photoionized positrons.

orders of magnitude lighter than a hydrogen atom, the velocity change associated with the absorption and emission of photons is concomitantly large. The recoil velocity $v_{\text{r}}$ due to a 243-nm-wavelength photon, which induces the 1*S*–2*P* transition, is approximately $v_{\text{r}} = 1.5 \times 10^3$ m s⁻¹ (equivalent to 55 mK). The large recoil velocity enables rapid deceleration. However, the corresponding change in resonance frequency due to the Doppler effect is 6.2 GHz, which is considerably larger than the natural linewidth of 50 MHz. Therefore, laser cooling halts after one cooling cycle if we use a narrow-bandwidth cooling laser, as commonly used in the Doppler cooling of ordinary atoms. Moreover, the recoil limit temperature is higher than the Doppler limit temperature, in contrast to ordinary atoms and molecules. The Doppler broadening of the 1*S*–2*P* transition at 300 K extends to approximately 460 GHz at the full-width at half-maximum (FWHM). In principle, the cooling of Ps is possible by using a laser pulse with a linewidth of the order of 100 GHz and a long duration, but the resulting temperature is limited to a few tens of kelvin[40].

Here, to overcome these challenges, we have extended the chirp cooling technique[41,42] commonly used to decelerate atomic beams. In chirp cooling, the frequency of the laser light changes over time to follow the change in the shifted resonance frequency due to deceleration, maintaining the cooling cycle. If the laser used has a very fast frequency chirp, which adapts to the large recoil-induced shift, substantial Doppler broadening and short lifetime, Ps can be cooled down to the recoil-limited velocity distribution, equivalent to sub-kelvin levels[41].

We used a previously demonstrated laser[43,44] that has the potential to realize such cooling. From this tailored laser, which is based on an injection-locked pulsed laser incorporating an electro-optic modulator in the laser cavity, short optical pulses of approximately 0.1 ns duration were successively output every 4.2 ns (Fig. 1a and Methods). Each pulse was spectrally continuous with a bandwidth of 8.9 GHz at FWHM, simultaneously covering all 1*S*–2*P* transition frequencies, triply split over 9.87 GHz. Thus, the pulse served for both cooling and repumping. The repumping prevents the reduction in cooling efficiency due to spin polarization of the 1*S* state during cooling (Methods). The central frequency of the pulse increases with each pulse, and the chirp is linear with a rate of $4.9 \times 10^2$ GHz μs⁻¹. The duration of the pulse train was adjustable up to approximately 1 μs. This chirped train of optical pulses is denoted here as the cooling laser. By effecting progressive deceleration from fast to slow Ps with chirp cooling in a counter-propagating configuration, we expect to obtain a velocity distribution close to the recoil-limited one. This laser cooling method is anticipated to produce slow Ps atoms in both electrostatic- and magnetostatic-field-free environments, marking an essential advancement in precision spectroscopy.

The experimental configuration inside the vacuum chamber is shown in Fig. 1b. Here we depict the pulsed generation of Ps gas in vacuum, three spatially superimposed laser beams and the acquisition of excitation signals for the 1*S*–2*P* transitions. The experiment was conducted at the Slow Positron Facility of the Institute of Materials Structure Science, the High Energy Accelerator Research Organization, Japan. Bunches of 10⁴ positrons (16 ns duration, 50 Hz repetition rate)[45] were guided

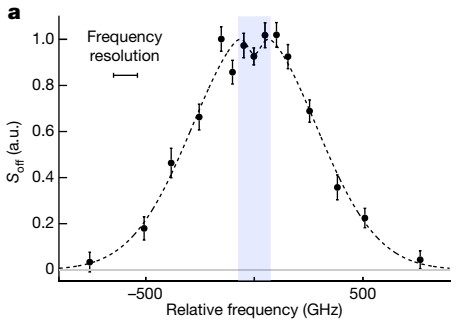

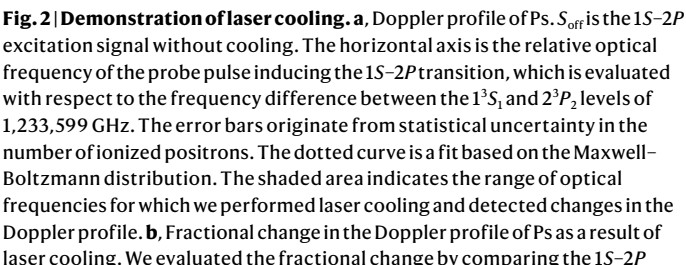

**Fig. 2 | Demonstration of laser cooling. a**, Doppler profile of Ps. $S_{off}$ is the $1S$–$2P$ excitation signal without cooling. The horizontal axis is the relative optical frequency of the probe pulse inducing the $1S$–$2P$ transition, which is evaluated with respect to the frequency difference between the $1^3S_1$ and $2^3P_2$ levels of 1,233,599 GHz. The error bars originate from statistical uncertainty in the number of ionized positrons. The dotted curve is a fit based on the Maxwell–Boltzmann distribution. The shaded area indicates the range of optical frequencies for which we performed laser cooling and detected changes in the Doppler profile. **b**, Fractional change in the Doppler profile of Ps as a result of laser cooling. We evaluated the fractional change by comparing the $1S$–$2P$

excitation signals with ($S_{on}$) and without ($S_{off}$) cooling laser irradiation. The filled circles represent the results when the optical frequencies of the cooling laser were set to a detuning suitable for cooling (swept from −59 to −9 GHz). The open circles represent the results of a control experiment with a larger detuning of the cooling laser (swept from −209 to −159 GHz). The dotted curve is a theoretical fitting to the data based on a phenomenological model (Methods). The spectral width of the probe pulse used for fit was 16 GHz. The dashed line is a constant function fitted to the data in a control experiment. In both cooling and control experiments, spectrally narrowed probe pulses were used.

through a magnetic field, focused by a magnetic lens and injected at room temperature into a sample of silica aerogel, which was used as the Ps formation medium[46]. Here γ-ray detection using scintillators facilitated the measurement of the arrival time of the positron bunch (Methods). Approximately $3 \times 10^3$ Ps atoms per positron bunch were released from the silica aerogel into the vacuum on the positron-injection side. Throughout this study, we evaluated the velocity distribution at 125 ns after production. At this point, we estimated that the Ps gas had a spatial spread of nearly 10 mm in the longitudinal and transverse directions, resulting in a density of approximately $10^3$ cm$^{-3}$. The three collimated laser beams encompassed the entirety of the Ps spatial spread.

The laser beams consisted of the cooling laser at 243 nm, a nanosecond laser pulse at 243 nm and a nanosecond laser pulse at 532 nm. We evaluated the velocity distribution of Ps by Doppler spectroscopy using the $1S$–$2P$ transition. Here we refer to the nanosecond laser pulse at 243 nm for Doppler spectroscopy as the probe pulse. The Ps atoms excited to the $2P$ state by the probe pulse were photoionized by the 532 nm pulse before relaxation to the $1S$ state. Photoionized positrons produced were collected by a microchannel plate (MCP) that was driven in a pulsed manner following the end of all the laser pulses, and the output current was converted to a voltage signal. This positron detection method offers high efficiency, enabling measurements to be conducted within a few days of the experimental beam time. Furthermore, the environment was electrostatic-field free until the interaction between Ps and laser pulses was complete. Figure 1c shows the timing chart from positron injection for Ps generation to the acquisition of the voltage signal from the MCP. The maximum magnitude of the residual static magnetic flux density was 0.15 mT over the entire experimental setup, indicating that its influence on Ps lifetime was negligible. In the following, the uncertainty of the signal level is dominated by the statistical Poisson uncertainty associated with the detected number of photoionized positrons.

Before proceeding with the laser cooling experiment, we examined the Ps gas temperature at 125 ns after the production of Ps from the Doppler profile (Fig. 2a). The measurement time for each frequency was approximately 20 min (equivalent to $1.2 \times 10^4$ measurement cycles; Methods). The profile has a spread around the theoretically known frequency difference between the $1^3S_1$ and $2^3P_2$ levels (1,233,599 GHz), reflecting the velocity distribution of Ps atoms in the $1S$ state. The gas temperature was estimated by assuming the Maxwell–Boltzmann distribution. Fitting the data with a model (Methods) that accounts for the 110 GHz spectral width of the multimode probe pulse and Lamb dip

associated with the counter-propagating laser beam revealed the gas temperature as $6.1(5) \times 10^2$ K (the source of the uncertainty is statistical). In aerogels with mean free paths of tens of nanometres[46], thermalization is relatively slow. If the Ps species generated by low-energy positrons are emitted into vacuum before fully thermalizing, this could account for the temperature difference between the aerogel and the Ps gas.

To demonstrate laser cooling, we irradiated the Ps gas with the cooling laser for 100 ns to maximize the number of atoms around zero velocity (Methods), commencing at the peak time of the positron pulse injection. The range of the Doppler shift of Ps affected by the cooling laser was approximately ±60 GHz with respect to the central frequency. Although it is possible to target the entire Doppler profile by extending the duration of the cooling laser, this is expected to yield fewer zero-velocity components (Methods). We measured the Doppler profile with an improved frequency resolution in the range mentioned above (Fig. 2a, shaded area) to best investigate the changes in the components interacting with the cooling laser. We obtained an optical frequency resolution ranging from 8 to 16 GHz by narrowing the 110 GHz linewidth of the original probe pulse using a Fabry–Pérot etalon (Methods). In the following, the measurement time for each frequency was 4 h, which is equivalent to $7.2 \times 10^4$ measurement cycles (Methods).

Figure 2b shows the fractional changes in the Doppler distribution. The excitation signal of the $1S$–$2P$ transition after irradiation with the cooling laser is defined as $S_{on}$ and the fractional change as $(S_{on} - S_{off})/S_{off}$. The filled circles show the results obtained when we tuned the optical frequency of the cooling laser to the conditions expected to produce a high cooling efficiency. The optical frequency (detuning) of the cooling laser was adjusted to chirp from approximately 1,233,540 GHz (−59 GHz) to 1,233,590 GHz (−9 GHz) during the 100 ns period.

The data show a decrease in the number of Ps in the optical frequency range swept by the cooling laser, accompanied by an approximately threefold increase in the number of Ps near zero velocity within a narrow velocity range, demonstrating the laser cooling of Ps—this result is the anticipated breakthrough achieved using our laser cooling technique. Such an increase in the slow component is essential for reducing systematic and statistical errors in the precision spectroscopy of short-lifetime systems.

We quantitatively evaluated the change in the Doppler profile resulting from laser cooling using a phenomenological model (Methods) to fit the measured data, taking into account the frequency resolution. The resultant fit is shown as the dotted curve in Fig. 2b. The best-fit value and the upper limit of the FWHM of the decelerated component were

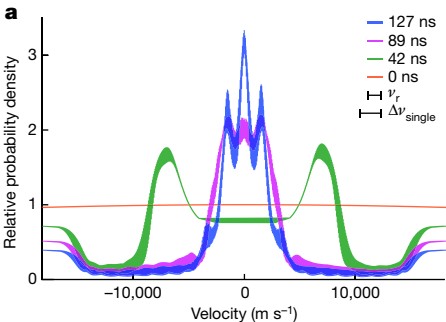

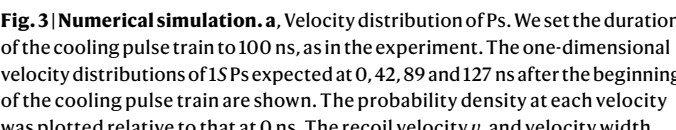

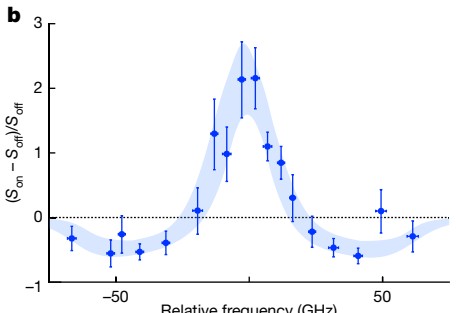

**Fig. 3 | Numerical simulation. a**, Velocity distribution of Ps. We set the duration of the cooling pulse train to 100 ns, as in the experiment. The one-dimensional velocity distributions of 1S Ps expected at 0, 42, 89 and 127 ns after the beginning of the cooling pulse train are shown. The probability density at each velocity was plotted relative to that at 0 ns. The recoil velocity $v_r$ and velocity width

$\Delta v_{single}$ (in resonance with the FWHM frequency width of a single pulse in the cooling laser) are also shown. **b**, Reconstructed fractional change in Doppler profile at 125 ns. The curve shows the simulated results. The thickness originates from the frequency resolution and uncertainty in the uncooled fraction. The filled circles are obtained from Fig. 2b.

23 and 30 GHz, respectively, which were evaluated by adopting the best resolution of 8 GHz. The corresponding temperatures were 0.8 and 1.4 K, indicating the cooling of velocities equivalent to approximately 1 K. In the frequency range swept by the cooling laser, reductions in the population of the 1S state were assessed to be 61% and 49%, respectively. Although the unknown number of delayed Ps released from the silica aerogel (Methods) precludes further quantitative discussion, most of the Ps resonating with the cooling laser were efficiently decelerated, as supported by the subsequent simulation (approximately 70% reduction in the swept frequency region, resulting in approximately 10% of the entire Ps population constituting the cooled component, assuming no delayed Ps release). These results, combined with the threefold enhancement of components near zero velocity, attest to the superior performance of our laser cooling technique. Considering that the unfixed frequency resolution varies between 8 and 16 GHz due to experiment-specific reasons (Methods), the aforementioned cooling performance, determined assuming the best frequency resolution, is probably an underestimate, as suggested by the numerical simulations presented below.

A control experiment was also performed in which the cooling laser was largely detuned. We tuned the optical frequency (detuning) to change from approximately 1,233,390 GHz (−209 GHz) to 1,233,440 GHz (−159 GHz) in 100 ns. In this case, Ps in the probed velocity range did not resonate with the cooling laser and no fractional change was expected. The open circles in Fig. 2b represent the fractional changes observed. The dashed line is a constant function fitted to the experimental results, indicating that there was no statistically significant change. Thus, the cooling laser does not impart a velocity change to the Ps that are off-resonant with it.

In Doppler spectroscopy, velocity distribution assessment is influenced by frequency resolution and 2P level splitting. To examine the presumed velocity distribution indicated in the experimental results, we conducted numerical simulations. We developed a framework to calculate the time evolution of the density matrix based on the Lindblad master equation (Methods). For the time–frequency characteristics and intensity of the cooling laser pulse train, we used parameters that were consistent with our experiments. Moreover, we did not include interatomic interactions and delayed Ps release from the aerogel.

Figure 3a shows how the velocity distribution of Ps in the 1S state changed over time with the cooling laser over a duration of 100 ns. The chirp rate of $4.9 \times 10^2$ GHz $\mu s^{-1}$ corresponds to a velocity change of 120 m $s^{-1}$ in 1 ns for Ps that resonate with the cooling laser. The single-pulse linewidth of 8.9 GHz (FWHM) corresponds to a velocity range of $\Delta v_{single} = 2,200$ m $s^{-1}$ (FWHM) that is covered.

The result at 42 ns (corresponding to a resonance velocity change of approximately 5,000 m $s^{-1}$) shows that Ps atoms with higher velocities

are sequentially decelerated without velocity intermittency and accumulate on the slower side. This corroborates the high deceleration efficiency of our cooling pulse train. At 89 ns (corresponding to a resonance velocity change of approximately 11,000 m $s^{-1}$), the decelerated Ps atoms were concentrated near zero velocity. The FWHM of the velocity distribution was approximately $3v_r$, and the corresponding temperature was 0.48 K.

At 127 ns, that is, 27 ns after the end of the cooling sequence, most of the excited Ps atoms had relaxed to the 1S state. Considering the frequency resolution and the possible delayed Ps release inferred from the fitting above, the simulated distribution quantitatively reproduced (Fig. 3b) the fractional change in the Doppler profile (Fig. 2b). In the velocity region where there was no interaction with the cooling laser, for example, near a velocity of 18,000 m $s^{-1}$, the number of Ps decayed by a factor of approximately 1/2.4 at 127 ns due to self-annihilation. However, the number of Ps near zero velocity increased by a factor of three or more compared with that at 0 ns.

Note that discrete peak structures were observed in the final velocity distribution. In addition to a distribution that shows an FWHM of approximately $v_r$ (equivalent to 55 mK) spread around the zero velocity, peaks that are shifted by $v_r$ from the zero-velocity component appeared. Although further study is needed, these structures indicate a sub-recoil cooling capability, similar to the dark states in Doppler cooling[47,48]. State-selective cooling in the final stage will be beneficial for the realization of a sub-100-mK Ps gas, eliminating the shifted peaks and transitions that induce residual acceleration.

We have successfully demonstrated the one-dimensional chirp cooling of Ps approaching the recoil limit. This innovative chirped cooling method can efficiently cool Ps to a hitherto unexplored narrow velocity range near zero, thereby providing scientific opportunities for this low-temperature, fundamental matter–antimatter bound system. Increased positron number and longer beam time will enable the verification of the sub-Doppler, narrow components indicated by simulations. Several strategies can be used to cool the entire Doppler distribution, such as extended cooling time with the presented chirp rate. The reduction in the number of cooled atoms indicated by our simulations (Methods) will be offset by increasing the slow positron beam intensity. The next natural extension is three-dimensional cooling (see Methods for the Doppler profile in the direction away from the surface of the aerogel; the current configuration did not provide cooling in this dimension), which reduces the second-order Doppler shift of the 1S–2S transition frequency by more than three orders of magnitude. Despite the two-component velocity distribution, the light mass of Ps aids the spatial separation of the cooled component with time allowing for selective excitation. However, there is a concern that three-dimensional cooling can result in higher temperatures or

lower decelerated populations than those reported here because the number of cooling cycles is limited by the annihilation lifetime. One solution is to use the conventional cooling method of thermal contact with low-temperature materials[18,49] as a precooling method. Enhancing the cooling rate by adopting stimulated emission, as used in atomic and molecular cooling[50,51], could also be beneficial. We believe that this more efficient three-dimensional cooling will enable the deceleration of most Doppler-broadened Ps atoms before annihilation, and is important for the realization of BEC in this particle–antiparticle system. The actual required density should be fully investigated because of the lower elastic scattering rates[52] between Ps at lower temperatures, which can markedly influence the thermalization for inherently short-lifetime systems. When achieved, comparisons with the BECs of excitons in semiconductors[53], another exotic atom system with a finite lifetime, will have important implications in quantum statistical physics.

A recent publication[54] submitted on the same date reports the one-dimensional cooling of Ps using the same transition as the present study. This experiment used the irradiation of a broadband, high-intensity cooling laser with a linewidth of approximately 100 GHz (r.m.s.) for a duration of 70 ns. This induced transitions in a broader velocity range of Ps than that in the present study, resulting in cooling from 380(20) to 170(20) K. In our research, we achieved the cooling of Ps over a velocity range equivalent to approximately 50 GHz by irradiating with a chirped train of pulses, each having a linewidth of 8.9 GHz, for 100 ns. Consequently, the Doppler broadening of the cooled components near zero velocity corresponds to approximately 1 K.

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

## Methods

### Ps generation

When a positron pulse is injected into certain media, some positrons form Ps and re-emit as Ps into vacuum. These Ps were used in this laser cooling experiment. Positron pulses, with a width of 16 ns and delivered at a repetition rate of 50 Hz (ref. 45), contained approximately $10^6$ positrons per pulse. The positrons were transported with an energy of 5 keV from the positron production unit to the experimental station guided by a typical magnetic field of approximately 10 mT generated by coils. By conducting current in the opposite direction only in the last coil, immediately before the experimental station, we minimized the magnetic field in the experimental region. Iron plates for magnetic shielding and the magnetic lens located downstream contributed to this minimization, further reducing the magnetic field in the experimental region to approximately 0.15 mT. This suppresses the Zeeman effect, resulting in a negligible annihilation rate of Ps in the $1^3S_1$ state (*ortho*-Ps) due to spin mixing. The transported positrons were focused onto the Ps formation medium using a magnetic lens. Approximately 1% of the incident positrons passed through the magnetic lens, and the remainder collided with the lens and annihilated. The instability in the number of positrons remained at approximately 1% throughout the experiment.

We used a silica aerogel, a three-dimensional network of $SiO_2$ (silica) nanograins, as the medium for the formation of Ps at room temperature. The silica aerogel had pores with a diameter of 45 nm and a porosity of approximately 95%. The incident angle of the positron bunch on the aerogel was 0° and approximately half the positrons injected into the silica aerogel formed Ps[46]. Some of the long-lived *ortho*-Ps atoms diffused and came out of the aerogel towards the experimental region. *Ortho*-Ps atoms decay into γ-rays with a vacuum lifetime of approximately 142 ns and these γ-rays were detected using a LaBr$_3$(Ce) scintillator and a plastic scintillator. The time-resolved γ-ray flux was measured by observing the current output of the coupled photomultiplier tubes. No influence was detected on the Ps from possible electrostatic charging of the silica aerogel.

### Cooling laser

A pulsed laser at 729 nm, which constitutes the backbone of the cooling laser, is called a chirped pulse-train generator[43]. This is an injection-locked pulsed laser equipped with an intracavity electro-optic phase modulator. It generates a train of approximately 0.1 ns pulses, each with progressively shifting central frequencies. These pulses are then amplified and frequency tripled, producing the 243 nm cooling laser light. The change in the central frequency over time (chirp) can be adjusted by changing the cavity length of the pulsed laser, the driving frequency and the modulation depth of the electro-optic modulator. The duration of the pulse train (the number of micropulses) can also be controlled up to approximately 600 ns, corresponding to a frequency sweep range of approximately 300 GHz using the chirp rate of the present study. Details on the operating principle of the chirped pulse-train generator can be found in ref. 43, and specifics regarding the design and performance evaluation of the cooling laser in this experiment are described in ref. 44.

Although the pulse duration has not yet been precisely measured, based on our measurements with an insufficient temporal resolution and the operating principle of the laser, a 0.1 ns pulse duration is estimated. Although an estimate (rather than a precise measurement), it does not affect the discussions in this paper as it is on a timescale considerably shorter than the natural lifetime of the 2P state.

### Laser configuration

Ps atoms emitted into vacuum were irradiated by three different pulsed lasers for chirp cooling in the one-dimensional direction as well as measuring the velocity distribution. The laser beams were incident in a direction orthogonal to the positron beam axis and reflected by a bare aluminium mirror in a counter-propagating configuration. The wavelengths of the light pulses used were 243 and 532 nm, with approximately 93% reflectance off the mirror at these wavelengths. The laser irradiation area was approximately 18 mm in the direction of the positron beam axis and approximately 8 mm in the vertical direction perpendicular to the positron beam axis.

The Ps-cooling chirped pulse-train laser was switched on for approximately 100 ns after the positron pulse impacted the silica aerogel. The fluence of a single pulse in an irradiating pulse train was typically 5 μJ cm$^{-2}$. During the irradiation period (approximately 100 ns), the central frequency of the light pulse was varied from 1,233,540 to 1,233,590 GHz, with a spectral width for a single pulse of 8.9 GHz (FWHM). The cooling laser was linearly polarized with a polarization direction orthogonal to the positron beam direction.

### Doppler spectroscopy

The velocity distribution of the *ortho*-Ps in the 1S state was evaluated by Doppler spectroscopy using the 1S–2P transition. The Doppler profile of the 1S state was obtained by measuring the signal associated with the number of positrons produced by ionizing the Ps in the 2P state as a function of the probe pulse frequency that resonantly induces the 1S–2P transition. The Doppler shift is indicative of the velocity of Ps along the propagation direction of the laser beam, allowing the evaluation of the *ortho*-Ps velocity distribution based on the measured Doppler profile. This measurement was conducted approximately 25 ns after the laser cooling ceased, after the complete de-excitation of the $2^3P_J$-state Ps through spontaneous emission. Doppler spectroscopy was carried out at a 10 Hz repetition rate, and the laser cooling occurred at a 5 Hz rate. By comparing the Doppler profiles before and after laser cooling, changes in the velocity distribution due to cooling can be assessed.

The second harmonic of an optical parametric oscillator (OPO) excited by the third harmonic of a $Q$-switched neodymium-doped yttrium aluminium garnet (Nd:YAG) laser was used as the probe laser for Doppler spectroscopy. The pulse duration was approximately 3 ns. The optical frequency of the probe laser was swept at approximately 1.2336 PHz and was measured using a wavelength meter with an accuracy of ±3 pm (corresponding to a frequency accuracy of approximately 15 GHz). Because of the longitudinal multimode nature, the spectral width of the second harmonic of the OPO was approximately $1.1 \times 10^2$ GHz. This spectral width was too wide to capture the changes in velocity profile resulting from chirp cooling. Therefore, the second harmonic of the OPO was transmitted through a solid etalon to narrow the spectrum and improve the velocity resolution. The measured transmission spectral width of our custom-made solid etalon available at 243 nm varied from 8 to 16 GHz at FWHM, depending on the angle and position of incidence.

However, the Doppler broadening of Ps without cooling has an FWHM of approximately $27\sqrt{T}$ GHz at a temperature of $T$ K. This corresponds to a frequency width of 470 GHz at room temperature, which is considerably wider than the narrow resolution. To measure the Doppler profile under uncooled conditions and evaluate the temperature, it was unnecessary to spectrally narrow the second harmonic of the OPO. A greater fraction of Ps with a distributed velocity was resonant, resulting in a larger signal. Therefore, we did not use the solid etalon when measuring the Doppler profile under uncooled conditions.

The typical incident fluences of the laser pulse that induced the 1S–2P transition were 0.27 and 2 μJ cm$^{-2}$ with and without spectral narrowing, respectively. This resulted in comparable light spectral densities for these two cases. The polarization of the laser pulse that induced the 1S–2P transition was linear and parallel to the positron beam.

For the ionization laser to photoionize Ps in the $2^3P_J$ state, we used the second harmonic (532 nm) of a $Q$-switched Nd:YAG laser with a pulse duration of 5 ns. This ionizing laser pulse was delivered with the timing

of the intensity peak adjusted to approximately 1.4 ns later than that of the ultraviolet nanosecond pulse, which induced the 1$S$–2$P$ transition. The irradiation fluence of the 532 nm pulse was typically 15 mJ cm$^{-2}$. We set the ionization laser to be linearly polarized parallel to the positron beam, similar to the laser that induced the 1$S$–2$P$ transition. The repetition rate of the ionizing laser was 10 Hz, same as that of the laser inducing the 1$S$–2$P$ transition.

Ionized positrons, produced with velocity selectivity from the Ps gas by the two-colour pulsed lasers, were drawn into an MCP. The MCP was placed immediately below the interaction region where Ps and the laser beams interacted. We applied a voltage of −2,000 V to the input surface of the MCP to collect the ionized positrons. The MCP was sensitive to the scattered photons of the deep-ultraviolet laser pulses at a wavelength of 243 nm, which resulted in a large background signal. Therefore, a pulsed negative voltage was applied to the MCP input surface after the completion of cooling laser irradiation, with a rise time of approximately 20 ns. Consequently, the MCP gain remained low at the time when the cooling and probe lasers were incident. This reduces the background signal originating from these photons, enabling the highly sensitive detection of ionized positrons. The voltage at the output plane of the MCP was set to 0 V. The amplified electrons were collected at a metal electrode, to which a constant voltage of 1,000 V was applied. The current output from this electrode was converted to a voltage with a 50 Ω resistor, and its time evolution was recorded. Positron signals were observed in the range of approximately 30–80 ns after the pulse voltage was applied, corresponding to the drift time that is dependent on the Ps location at photoionization. Although the background signal originating from the deep-ultraviolet photons was substantially reduced, a residual signal remained. To subtract this contribution, the ionizing laser was switched on and off every 30 s and we evaluated the signal of ionized positrons based on the difference in the integrated signals of the MCP with and without the ionizing laser.

When the number of Ps in the whole velocity distribution was approximately 3 × 10$^3$ immediately after production, the average number of detected positrons in the frequency-resolved measurements was typically 0.5. Consequently, the uncertainty in the excitation signal in the Doppler spectroscopy measurements was characterized by the randomness in the number of ionized positrons, which is governed by Poisson statistics. To achieve an adequate signal-to-noise ratio, it was necessary to set an appropriate measurement time. For the Doppler spectroscopy used to assess the temperature of Ps gas with a frequency resolution of 110 GHz, the integration time for each probe frequency was approximately 20 min. During this integration time, the number of measurement cycles was approximately 1.2 × 10$^4$. In the laser cooling experiment, for which the resolution was set to be an order of magnitude greater and thus the signal was weaker, the integration time was approximately 4 h for each probe frequency. During this time, the number of measurement cycles was approximately 7.2 × 10$^4$.

### Analysis of the measured Doppler profile of uncooled Ps

We estimated the temperature of the Ps emitted from the silica aerogel using the measured Doppler profile. For this purpose, we defined a model function, which was fitted to the data. The model function describes the number of positrons $S(\omega_R)$ generated by the photoionization process from the 2$P$ state as a function of the central angular frequency $\omega_R$ of the probe pulse that induces the 1$S$–2$P$ transition. $S(\omega_R)$ is written as

$$S(\omega_R) = \int D(v; T) \frac{C}{1 + \frac{I_S}{I(v; \omega_R)}} dv,$$

where $D(v; T)$ is the probability density of Ps with velocity $v$ and temperature $T$ (the Maxwell–Boltzmann distribution function is used); $I_S$

is the saturation intensity at the 1$S$–2$P$ transition angular frequency $\omega_{eg}$; $I(v; \omega_R)$ is the light intensity at the angular frequency resonant to a Ps atom with velocity $v$; and $C$ is a constant and free parameter in the fitting. The second term in the integral represents the photoionization probability of Ps at velocity $v$. The functional form for $S(\omega_R)$ was determined using the following relation[55]:

$$P_e = \frac{1}{2\left(1 + \frac{I_S}{I_R}\right)},$$

which describes the occupation probability of the excited state in a two-level system when irradiated with light of the transition frequency at intensity $I_R$ (see the denominator in the fraction). We used a two-level approximation because we set the spectral width of the probe pulse to be sufficiently wide compared with the splitting in the 1$S$–2$P$ transition frequency. $S(\omega_R)$, determined using $P_e$, describes the nonlinear responses to the probe laser pulse, such as the Lamb dip and saturation broadening effects in the present Doppler-broadened case.

In our experiment, we directed each laser beam at the Ps in a counter-propagating configuration. Therefore,

$$I(v; \omega_R) = I_L\left(\omega_{eg} + \frac{v}{c}\omega_R; \omega_R\right) + I_L\left(\omega_{eg} - \frac{v}{c}\omega_R; \omega_R\right),$$

where $I_L(\omega; \omega_R)$ is the intensity spectrum, described as a function of $\omega$, of the probe pulse with its central angular frequency $\omega_R$, and $c$ is the speed of light. We adopted the measured spectral width of $I_L(\omega; \omega_R)$, with the intensity being a free parameter in the fitting. Here we did not include the spatial distributions of light intensity and Ps density. The light intensity of the probe pulse that reproduced the measurement was consistent with the actual light intensity calculated using the fluence, pulse duration and spectral width. This result demonstrates the validity of the proposed model.

The Doppler profile in the direction normal to the surface of the silica aerogel (Extended Data Fig. 1) was measured by the single-path irradiation of the probe pulse and ionization pulse. These optical pulses (diameter, approximately 10 mm) propagated towards the aerogel 125 ns after the peak timing of the positron bunch. The angles of incidence on the aerogel were 0° for the probe beam and 22° for the positron bunch. The peak of the Doppler profile was observed at a relative frequency of approximately −360 GHz, with an FWHM of approximately 390 GHz. In contrast to the velocity components parallel to the aerogel surface, which are randomly distributed, the distribution of the velocity components perpendicular to the surface cannot be effectively described by a simple distribution function that represents gases or beams. The velocity of Ps moving away from the surface depends not only on its velocity in the generating material but also on its work function. Consequently, the velocity distribution can generally differ from the component parallel to the surface. For these reasons, we did not perform an evaluation by fitting the experimental data. Irrespective of the parallel or perpendicular direction to the surface, the emission velocity of Ps changes dynamically with the reduction in momentum due to scattering with the molecules comprising the aerogel, with the velocity distribution also being influenced by the decay due to the Ps lifetime.

### Analysis of fractional change in Doppler profile using a phenomenological model

We analysed the fractional change in the velocity distribution induced by the cooling laser by fitting the following phenomenological model to the data: the fractional change for $S_{on}(f)$ and $S_{off}(f)$ is defined as

$$\frac{S_{on}(f) - S_{off}(f)}{S_{off}(f)}.$$

We first used the following raw functions that did not include the frequency resolution in the experiment:

$$
S_{\text{on}}^{\text{raw}}(f) =
\begin{cases}
\exp\left(-\dfrac{m_{\text{Ps}}\,c^2 f^2}{2k_{\text{B}}T_0 f_0^2}\right), & f < -f_{\text{cooled}},\ f_{\text{cooled}} < f \\[2mm]
A\exp\left(-\dfrac{4\log 2\,f^2}{\Delta f^2}\right) + S_{\text{cooled}}, & -f_{\text{cooled}} \le f \le f_{\text{cooled}}
\end{cases}
$$

$$
S_{\text{off}}^{\text{raw}}(f) = \exp\left(-\frac{m_{\text{Ps}}\,c^2 f^2}{2k_{\text{B}}T_0 f_0^2}\right),
$$

where the argument $f$ is the relative frequency, $f_0$ is the $1^3S_1$–$2^3P_2$ transition frequency of Ps, $m_{\text{Ps}}$ is the mass of Ps, $k_{\text{B}}$ is the Boltzmann constant and $T_0$ is the temperature of Ps released from the silica aerogel. On the basis of the experimental results, we assumed that the Doppler profile of the uncooled Ps was a Maxwell–Boltzmann distribution at temperature $T_0 = 600$ K. The following free parameters used in the fit describe the change in Doppler profile associated with cooling: $f_{\text{cooled}}$ is the Doppler shift corresponding to the optical frequency at the beginning of the cooling laser; $\Delta f$ is the Doppler width of the decelerated component; $S_{\text{cooled}}$ is the signal level after cooling in the spectral region swept by the chirped cooling laser; $A$ characterizes the magnitude of the decelerated component signal. These raw functions are plotted in Extended Data Fig. 2.

We generated model functions $S_{\text{on}}(f)$ and $S_{\text{off}}(f)$, which correspond to the experimental results obtained, by convolving $S_{\text{on}}^{\text{raw}}(f)$ and $S_{\text{off}}^{\text{raw}}(f)$ with the frequency resolution due to the linewidth of the probe pulse. The change in Doppler profile associated with cooling was quantitatively evaluated by fitting the modelled fractional change to the measured fractional changes. In Extended Data Fig. 2, $S_{\text{on}}^{\text{raw}}(f)$ is plotted using the parameters obtained from the fit.

The fitting parameters varied with the spectral width of the probe pulse, which determined the frequency resolution of the measured Doppler profile. When the spectral width, which varied in the experiment, was set to 8 GHz (narrowest), the widest Doppler spread of the cooling component was evaluated. In the main text, we have shown the corresponding best-fit value (23 GHz) and upper statistical limit (30 GHz) as conservative estimates (the upper statistical limit of the width of the cooled component was evaluated at the 95% confidence level). The population reductions in the cooled spectral region were estimated to be 61% and 49%, respectively. For a spectral width of 16 GHz for the probe pulse, the best-fit value and the upper limit of the width of the cooled component were 18 and 27 GHz, and the corresponding population reductions were 78% and 61%, respectively.

We believe that the estimated population reductions were smaller than those expected from population reduction by laser cooling alone, due to the influence of delayed Ps release from the silica aerogel. Such delayed release from porous materials has been reported previously[56]. Our empirical observations suggest the presence of Ps emitted from the silica aerogel several tens of nanoseconds after the injection of the positron bunch, when we observed the components of zero lateral velocity. However, we cannot quantitatively discuss the delayed fraction due to the absence of available systematic data.

## Evaluation of the frequency resolution in the laser cooling experiment

The frequency resolution of the fractional change in the Doppler profile as a result of laser cooling was determined using the spectral width of the probe pulse and intensity-dependent saturation broadening. We evaluated the spectral width of the probe pulse using the optical resolution of the Fabry–Pérot solid etalon used for spectral narrowing. The FWHM optical frequency resolution as a function of the angle of incidence is shown in Extended Data Fig. 3. The resolution was evaluated by measuring the transmission spectrum of single-longitudinal-mode laser pulses at 243 nm. The spectral width of the pulses is expected to be less than 10 MHz, which is considerably narrower than the designed frequency resolution of the solid etalon, thereby enabling the evaluation of the actual resolution. We measured the transmittance as a function of the angle of incidence of the etalon. All the incident angle sweeps designated in the legend were performed in the direction of increasing angle. These three sets of measurements were performed in the experimental period but not consecutively.

The results indicate that although the transmission spectral width tends to increase with the angle of incidence, it varies widely for each measurement. The degree of variation exceeds the measurement uncertainty, suggesting that the conditions of the etalon changed with each sweep. The possible characteristics of the solid etalon that can cause such variations include non-uniform thickness and inhomogeneous strain on the etalon. Variations can then occur because the position of the laser irradiation on the solid etalon cannot be completely fixed. To detect a change in the Doppler profile resulting from laser cooling, the angles of incidence of the probe pulse on the etalon were set in the range tested above, resulting in the same degree of variation in the linewidth of the probe. Therefore, we estimated the spectral width of the probe pulse to be 8–16 GHz, based on the measured range of values shown in Extended Data Fig. 3.

Next, we examined the influence of saturation broadening, which also affects the frequency resolution. Using the effective intensity calculated from the fluence, pulse duration and spectral width of the spectrally narrowed probe pulse, the degradation of the frequency resolution owing to saturation broadening was at most 1 GHz. Thus, saturation broadening can be neglected.

We considered the 8–16 GHz range of the frequency resolution as a systematic uncertainty in the evaluation of the fractional change. Hence, a conservative effective temperature was evaluated.

## Allowed $1^3S$–$2^3P$ transitions and their intensities

Here we describe the allowed transitions and their intensities among the $1^3S$–$2^3P$ transitions used for laser cooling and Doppler spectroscopy. The transition matrix element is

$$
-\langle n=2, L=1, S=1, J_{\text{e}}, M_{\text{e}}\,|\,\mathbf{d}\,|\,n=1, L=0, S=1, J_{\text{g}}, M_{\text{g}}\rangle \cdot \mathbf{E},
$$

where $\mathbf{d}$ is the electric dipole moment; $\mathbf{E}$ is the electric field of light; $n$, $L$ and $S$ are the principal quantum number, orbital angular momentum and total spin angular momentum, respectively; $J$ and $M$ are the total angular momentum and its projection along the quantization axis, respectively. Subscripts e and g indicate the excited and ground states, respectively.

The electric dipole moments, when we define the quantization axis of the atomic orbitals as the $z$ axis, are shown in Extended Data Fig. 4a,b. The direction of projection of the electric dipole moment is shown at the top of each diagram. The allowed transitions induced by the electric field of light with the corresponding polarization vectors are represented by arrows. The numbers associated with the arrows indicate the square of the absolute value of each component of the electric dipole moment normalized to the following constant:

$$
|d_0|^2 = \left(\frac{128\sqrt{2}}{243}2ea_0\right)^2,
$$

where $e$ is the elementary charge and $a_0$ is the Bohr radius. For some transitions, the numbers are omitted because the absolute values of the

electric dipole moments coincide with those of the other transitions that differ only in the sign of $M$. The transition rates are proportional to the values shown in Extended Data Fig. 4a for linearly polarized light parallel to the $z$ axis, and in Extended Data Fig. 4b for orthogonally polarized light. In our experiment, the polarizations of the cooling laser pulse and probe laser pulse in Doppler spectroscopy were linear and orthogonal to each other. Extended Data Fig. 4a,b can be used to evaluate the transition intensity of each pulse.

Extended Data Fig. 4c shows the spontaneous emission rates from the excited states to each ground state normalized by the total decay rate $\Gamma_{sp.} \cong 3.13 \times 10^8\,s^{-1}$ from each excited state. By symmetry, the spontaneous emission rates from the states with negative $M_e$ values, which are omitted from the table in Extended Data Fig. 4c, are equal to the corresponding rates between the states with the signs for $M_e$ and $M_g$ reversed.

Extended Data Fig. 4a–c shows that in the cooling process, during which the transitions are repeated many times, it is important to use a cooling laser with a spectral width comparable with the splitting in the transition. Otherwise, if we repeat the cooling cycle by transitioning to the $2^3P_0$ and $2^3P_1$ states, for example, the $1^3S_1$ state becomes polarized and eventually makes transitions to these excited states dark. Moreover, the $1^3S_1$–$2^3P_2$ transition dominated the $1S$–$2P$ transitions. Therefore, we present our experimental results as functions of frequency relative to the $1^3S_1$–$2^3P_2$ frequency difference. Note that the resonance frequency observed at the one-photon transition is approximately 3 GHz higher than this frequency difference owing to the conservation laws of energy and momentum.

#### Numerical simulation

We evaluated the time evolution of the momentum distribution of Ps under the influence of a cooling laser based on the Lindblad master equation:

$$\frac{d\rho}{dt} = \frac{1}{i\hbar}[H,\rho] + L(\rho),$$

where $t$, $\hbar$, $H$ and $L(\rho)$ are the time, Dirac's constant, Hamiltonian and Liouvillian, respectively. We considered the density matrix $\rho$ in the space spanned by the simultaneous eigenstates of the momentum of Ps and atomic configurations in the $L$–$S$ coupling scheme. The interaction between Ps and the photon field was incorporated as an electric dipole interaction. This framework can describe the transitions between atomic orbitals through absorption, stimulated emission and spontaneous emission processes, as well as momentum changes because of photon recoil. We incorporated the relaxation of Ps due to annihilation processes into the master equation as a longitudinal relaxation process.

Using the simulated velocity distribution shown in Fig. 3a, we can simulate the fractional change (Fig. 2b). The simulated Doppler profiles with and without the cooling laser irradiation were convolved by the spectral resolution to obtain $S_{on}^{sim}(f)$ and $S_{off}^{sim}(f)$, respectively. The spectral width of the probe pulse determines the spectral resolution. The argument $f$ is the relative frequency, which is the first-order Doppler shift calculated from the velocity of Ps. To express a part of the probed Ps atoms, which interacted with the cooling laser, we introduce an uncooled Ps fraction $r$. The fractional change can then be calculated as $(1-r)\frac{S_{on}^{sim}(f) - S_{off}^{sim}(f)}{S_{off}^{sim}(f)}$. The parameter $r$ was determined by fitting this function to the measured data. Figure 3b compares the measured and simulated fractional changes. The filled circles are identical to those shown in Fig. 2b. The thickness of the curve was determined on the basis of the frequency resolution in the range of 8–16 GHz. The measured data were well reproduced, with the best estimated $r$ ranging from 0.18 to 0.40 and the spectral resolution shown above. The statistical uncertainty of the estimated $r$ is typically 0.06 at the $1\sigma$ confidence level. The resultant fraction $r$ is reasonable under the experimental condition, and its consistency with the measured data supports the successful demonstration of the laser cooling of Ps.

Extended Data Fig. 5 presents the corresponding Doppler profiles mentioned above. Similar to the experiment, the simulation evaluated the Doppler profiles 125 ns after Ps formation, following 100 ns of cooling laser irradiation. Components resonating with the frequency-swept cooling laser were decelerated and concentrated in the frequency domain corresponding to zero velocity. Compared with the case without cooling, the slow components showed a threefold increase. No change was observed in the detuned components, which did not resonate with the cooling laser.

To illustrate the parameter design of the cooling laser in this study, we present typical examples of cooling time dependence and chirp rate dependence based on the simulations constructed here. Extended Data Fig. 6a displays the momentum distribution after cooling, evaluated as a function of cooling time while keeping the chirp rate constant. The optical frequency detuning of the cooling laser at the end of cooling was set to −9 GHz. As the cooling duration is extended, the sweep frequency range of the cooling laser increases, thereby enhancing the contrast in number between the cooled and uncooled components. However, due to the lifetime effects of Ps, the number of cooled atoms is found to decrease compared with shorter cooling times. The maximum number of cooled atoms is achieved at a cooling time of approximately 100 ns, which is the duration used in this study.

Extended Data Fig. 6b shows the momentum distribution after cooling when the chirp rate is varied, with the cooling time fixed at 100 ns. When the chirp rate exceeds the rate characterized by the recoil frequency associated with photon absorption and the natural emission rate, the proportion of atoms that cannot maintain the chirp cooling cycle increases, resulting in decreased efficiency. Note that this calculation assumes that the entire volume of the Ps gas is constantly exposed to the cooling laser.

#### Data availability
The data used in the current study are available on Figshare at https://doi.org/10.6084/m9.figshare.26316208 (ref. 57).

#### Code availability
The codes used for modelling or analysis in the current study are available from K. Yoshioka on request.

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

**Acknowledgements** This study was supported by the MEXT Quantum Leap Flagship Program (MEXT Q-LEAP) grant no. JPMXS0118067246; JST FOREST Program (grant no. JPMJFR202L); JSPS KAKENHI grant nos. JP16H04526, JP17H02820, JP17H06205, JP17J03691, JP18H03855, JP19H01923, JP21K13862, JP22KJ0637 and JP24H00217; Grant for Basic Science Research Projects from The Sumitomo Foundation, MATSUO FOUNDATION, Mitutoyo Association for Science and Technology (MAST); Research Foundation for Opto-Science and Technology; and Mitsubishi Foundation. This study was performed under the approval of the Photon Factory Program Advisory Committee (Proposal nos. 2020G101, 2022G087 and 2023G660). This study used the Fugaku computational resources provided by the RIKEN Centre for Computational Science through the HPCI System Research Project (project ID hp230215). Y.T. and R.U. acknowledge support from FoPM, and K. Yamada acknowledges support from XPS, WINGS Programs of The University of Tokyo. We thank S. Uetake and K. Yoshimura (Okayama University) for providing the OPO; R. Suzuki for technical suggestions regarding the experimental setup; T. Hatae and M. Hayashi for cooperation regarding the nanosecond lasers; and H. Katori (The University of Tokyo), M. Kuwata-Gonokami (RIKEN), E. Chae (Korea University) and N. Zafar for discussions.

**Author contributions** K. Yoshioka directed the laser cooling experiments. The cooling laser was conceived by K. Yoshioka, with essential updates to the design and subsequent construction provided by K.S. The experiments were designed and performed by K.S., Y.T., R.U., N.M. and K. Yoshioka. The laser pulses were characterized by K.S., N.M. and K. Yoshioka. The laser system for Doppler spectroscopy in the early stages of the experiment that detected γ-rays was developed by K. Yamada, and was extended with highly sensitive detection using an MCP by K.S., Y.T. and K. Yoshioka. Data were analysed by K.S., N.M., S.S. and K. Yoshioka. The data

acquisition program was developed by K.S., A.I., S.S. and T.K. The numerical simulation was developed by K.S., R.U., S.S. and K. Yoshioka. A.I. led the preparation of Ps atoms. The development of γ-ray detection systems and the analysis of γ-ray data were performed by A.I., R.W.G., T.N. and S.A. Positron beam alignment was conducted by K.W., I.M. and T.H. Evaluation of the silica aerogel was performed by K.S., A.I., K.I., K.M., B.E.O. and N.O. The magnetic lens was provided by N.O. A.I., T.N., R.W.G., N.O., K.W. and T.H. focused the positron bunch on the target aerogel and contributed to the reduction in static magnetic field in the interaction region. A.I. and R.W.G. performed a blind analysis of the laser cooling data. K.S. and K. Yoshioka wrote the manuscript with feedback from all the authors.

**Competing interests** The authors declare no competing interests.

**Additional information**
**Correspondence and requests for materials** should be addressed to A. Ishida or K. Yoshioka.

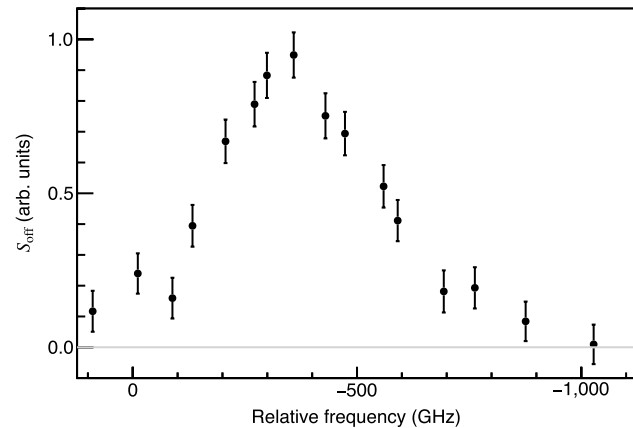

**Extended Data Fig. 1 | Doppler profile normal to the aerogel surface.** The profile was measured at 125 ns after Ps production, using a single-pass probe beam directed towards the aerogel. The negative detuning indicates the direction away from the surface and the horizontal axis is intentionally reversed.

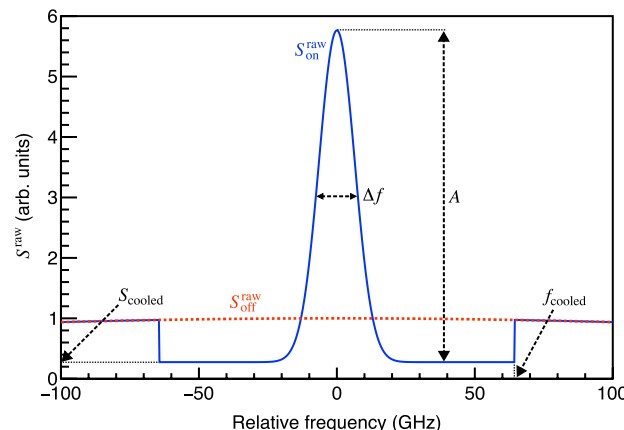

**Extended Data Fig. 2 | Raw Doppler profile functions used to analyse the fractional change in velocity distribution associated with laser cooling.** The Doppler distribution of uncooled Ps follows the Maxwell–Boltzmann distribution, and the possible changes in the Doppler profile by cooling are characterised by parameters $A$, $S_{\text{cooled}}$, $\Delta f$, and $f_{\text{cooled}}$ (defined in the text). Fitting of the experimental data was performed by considering the frequency resolution in Doppler spectroscopy.

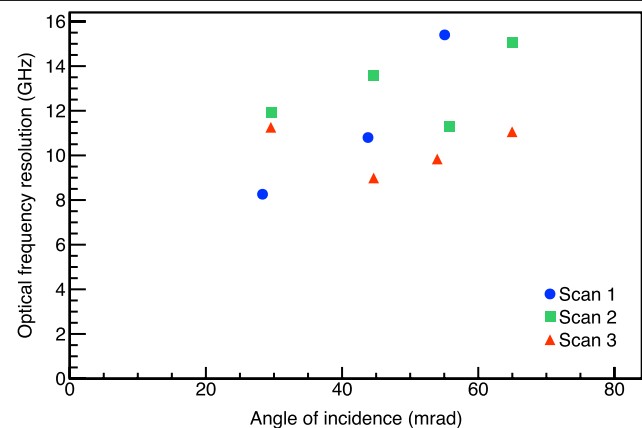

**Extended Data Fig. 3 | Optical frequency resolution of the solid etalon used in the spectral narrowing of the probe pulse for the 1S-2P transition.** We evaluated the FWHM resolution as a function of the incident angle of single-longitudinal-mode pulses. Differences in the markers represent the three scanning experiments.

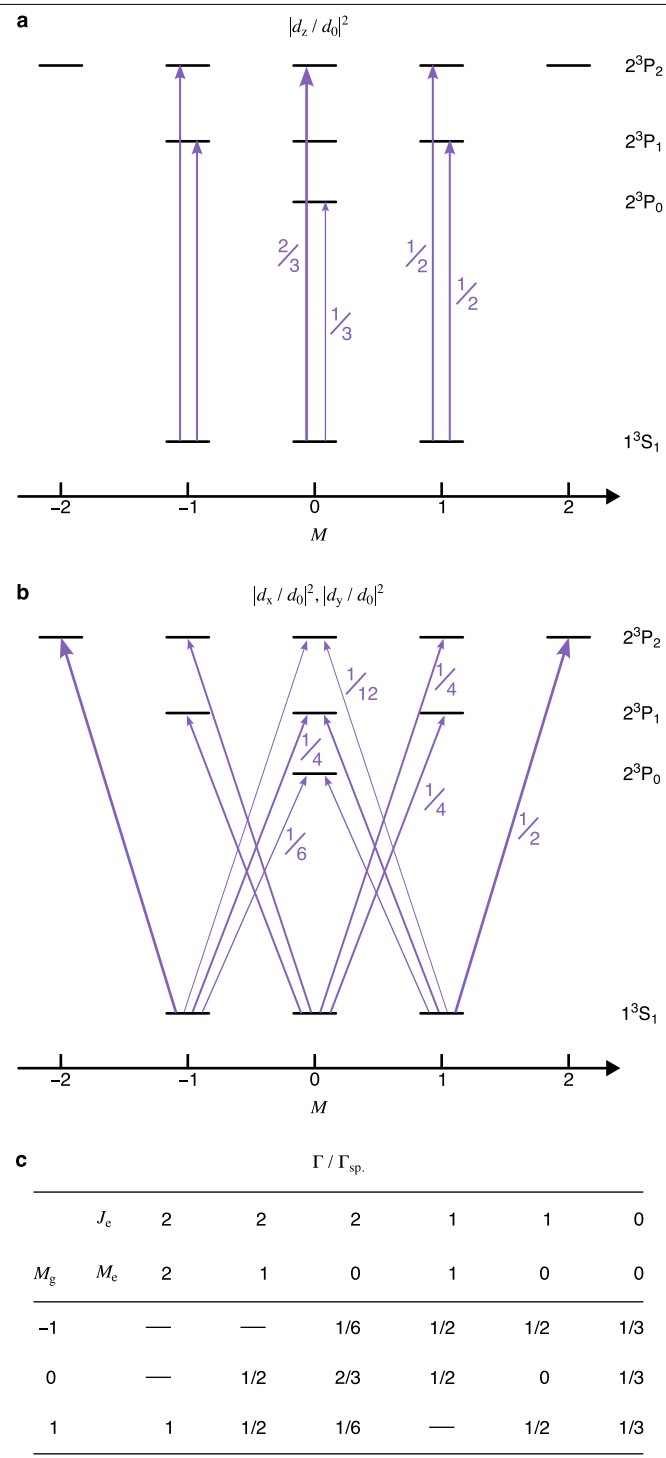

**a** $|d_z / d_0|^2$

2³P₂
2³P₁
2³P₀

²⁄₃   ½   ½   ½

⅓

1³S₁

−2   −1   0   1   2
$M$

**b** $|d_x / d_0|^2, |d_y / d_0|^2$

2³P₂
2³P₁
2³P₀

1/12   ¼

¼   ¼

⅙   ½

1³S₁

−2   −1   0   1   2
$M$

**c** $\Gamma / \Gamma_{sp.}$

| $M_g$ | $J_e$ | 2 | 2 | 2 | 1 | 1 | 0 |
|---|---|---|---|---|---|---|---|
| | $M_e$ | 2 | 1 | 0 | 1 | 0 | 0 |
| −1 | | — | — | 1/6 | 1/2 | 1/2 | 1/3 |
| 0 | | — | 1/2 | 2/3 | 1/2 | 0 | 1/3 |
| 1 | 1 | 1/2 | 1/6 | — | 1/2 | 1/3 |

**Extended Data Fig. 4 | Allowed 1³S–2³P transitions and their intensities.**
**a**, **b**, Magnitudes of the electric dipole moments. The subscripts x, y, and z of the variable $d$ at the top of each figure refer to the direction of projection of the dipole moment. Numbers attached to the arrows represent the squared ratio of the projected dipole moment with respect to the constant $d_0$ defined in the text. $M$ refers to the projection of the total angular momentum $J$ onto the quantisation axis. **c**, Summary of spontaneous emission rates. Numbers in the table cells represent the ratio of the spontaneous emission rate from the excited state with quantum numbers $J_e$ and $M_e$ to the ground state with $M_g$. $\Gamma_{sp.}$ denotes the total spontaneous emission rate from each excited state.

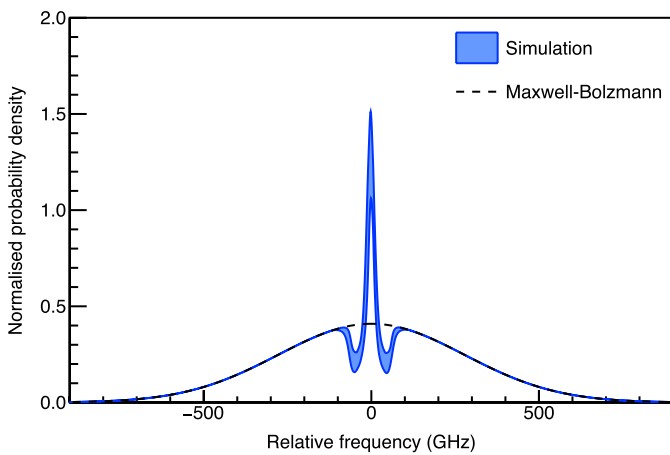

**Extended Data Fig. 5 | Simulated Doppler profile resulting from the laser cooling in the present experiment.** The curve shows the simulated result 125 ns after the production of Ps. The uncertainty in the spectral width of the probe pulse and the resultant uncertainty in assessing the non-cooled fraction contribute to the thickness of the curve. The dotted curve is the calculated Doppler profile at 600 K without laser cooling. Both curves are normalised with respect to the Doppler profile at 0 ns without laser cooling.

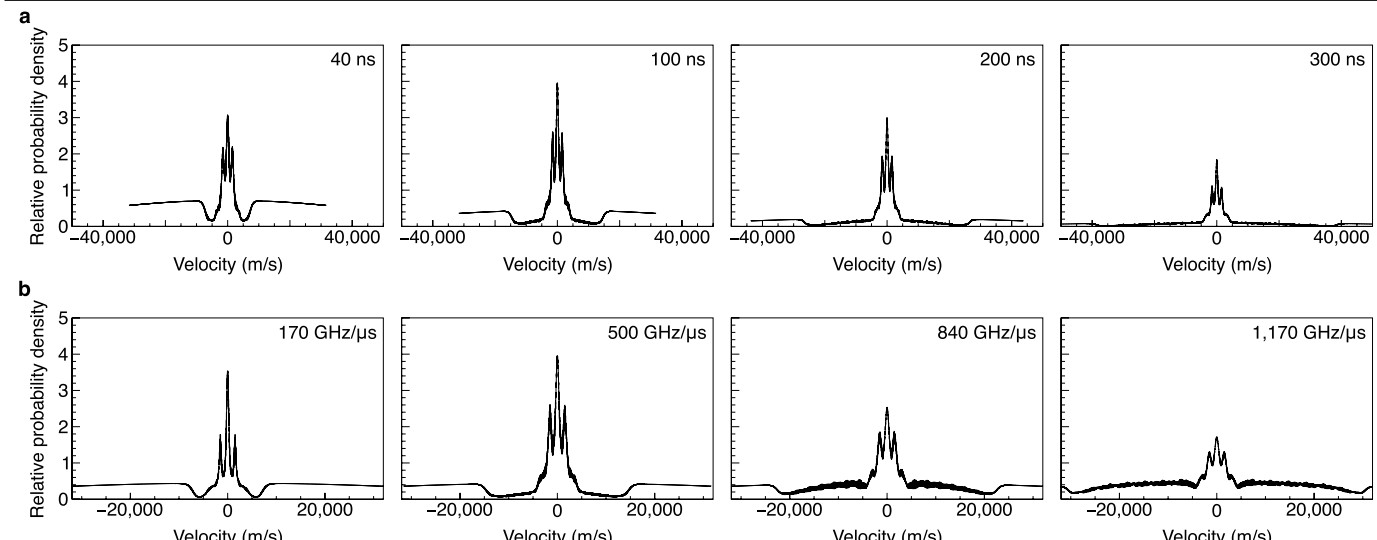

**Extended Data Fig. 6 | Simulation results showing the velocity distribution of the 1 S state 25 ns after the end of the cooling laser irradiation, relative to the set parameters of the cooling laser.** The probability density is plotted relative to the maximum value of the velocity distribution at a temperature of 600 K at time zero. Irradiation by the cooling laser commenced at time zero. **a**, Dependence on the duration of the cooling laser. The chirp rate of the up-chirping cooling laser was the same as in the experiments of the present work, at $4.9 \times 10^2$ GHz/µs, and cooling was terminated at −9 GHz relative to the 1S-2P transition frequency at rest. The durations were 40 ns, 100 ns, 200 ns and 400 ns. Due to computational resource constraints, calculation of the largely detuned components was truncated. **b**, Dependence on the chirp rate of the cooling laser. We set the chirp rates at $1.7 \times 10^2$ GHz/µs, $5.0 \times 10^2$ GHz/µs, $8.4 \times 10^2$ GHz/µs and $11.7 \times 10^2$ GHz/µs. Cooling was terminated at −9 GHz relative to the stationary Ps 1S-2P transition frequency. The duration of the cooling laser was fixed at 100 ns. While a larger chirp rate allows for interaction with a broader velocity range, the efficiency of cooling decreases because the frequency of the cooling laser changes by more than the recoil frequency during the relaxation time due to natural emission from the 2 P state.