## [Peer Review file · Nature]

Manuscript Title: Cooling positronium to ultra-low velocities with a chirped laser pulse train

Reviewer Comments & Author Rebuttals

Reviewer Reports on the Initial Version:

Referees' comments:

Referee #1 (Remarks to the Author):

This is the first exhibition of the partial laser cooling of positronium. Could be useful for precision measurements of the 1S-2S interval of positronium, maybe not so useful for making a BEC.

Referee #2 (Remarks to the Author):

This article reports one-dimensional laser cooling of positronium using an innovative frequency-chirped, pulsed laser. The authors demonstration of laser cooling involves exposing a pulsed positronium sample to the chirped laser for 100 ns. A probe setup consists of a pulsed, 243 nm laser to drive the 1S-2P transition, and a nanosecond-pulsed laser at 532 nm to ionise the 2P state. The positrons released in the ionisation are counted on a multichannel plate. The evidence for laser cooling is a single figure in which an uncooled spectrum is compared to a laser-cooled spectrum. The authors have examined only the central region of the Doppler profile for the cooled sample, and their claim is based upon increase of the signal over a narrow frequency range around zero detuning. A corresponding decrease in signal just outside of the enhanced region is presumably due to atoms that have been cooled towards the central region. The authors also present a simulation of the process in Figure 3.

This article represents a technical tour-de-force in the preparation and manipulation of positronium. The methods appear to be valid and original. However, I hesitate to recommend publication in Nature at this stage because of the small amount of evidence presented compared to the importance of the claim. In particular, the focus on only the central region of the velocity distribution in Figure 2b is concerning. It is clear that the effects of cooling should be visible in this region, but why not probe the whole profile, as in Figure 2a? This is, in my opinion, a serious oversight. Figure 2b shows a signal enhancement of a factor of more than two at zero relative frequency. This would be clearly visible on a broad scan, and the reader wouldn't be left wondering what happened to the rest of the distribution. A zoomed-in figure as shown could also be added.

I am also missing a description of how such a two-component velocity distribution would be useful in future experiments. Surely the promise of laser cooling is fully realised when the whole distribution is cooled. The authors don't comment on this or outline how to achieve it in the future. I am also missing - in the main text - some quantitative measure - with uncertainties - of the efficacy of the laser cooling. Cooled temperatures of 0.8 and 1.4K are quoted, but the nature of the uncertainties in this article is generally hard to find.

While this work is exciting and innovative, I am not yet convinced that it is advanced enough to live up to the claim of the first laser cooling of positronium. We are also missing any characterisation of the degrees of freedom perpendicular to the cooling laser. It would also be good, but perhaps not absolutely necessary, to see another spectrum where the authors have varied something - say detuning or chirp rate - and seen an expected response.

I have added some detailed comments below. They should be addressed in any future versions of the submission.

In the following, I refer to the line numbers in the Nature-provided PDF file.

25. What does 'direction independent' mean? Please revise.

39. It is not clear that progress in this field has anything to do with matter-antimatter asymmetry. This is a motivation, but there is as yet no obvious link.

43. reference needed for BEC of positronium

48-49. Be quantitative about the current precision for experiment and theory.

54. What about antiprotonic helium? What does 'a satisfactory level' mean

56. What does high temperature mean here? You should be very explicit for the Nature readership.

66-73. This description of laser cooling is not very illuminating, and it applies only to systems at rest in the lab. There is no mention of the Doppler effect and laser detuning here. Zeeman slowing is not cooling.

80. It would be helpful to make a temperature/velocity comparison here for positronium. E.G. what is the recoil velocity in equivalent temperature units.

82. You should explicitly state the natural linewidth here. this is key to all of the future discussion.

83. I think you mean 'narrow bandwidth' and not single frequency here.

89. Is there a reference needed here for the 10K limit?

98. There is a reference to a 'tailored laser (see Fig 1a)', but there is nothing about the laser in the figure except the pulse spectra. I was expecting a depiction of the laser system.

98. 'short optical pulses' is not helpful; be quantitative

100. The frequency widths between the three 2P levels (4.4 and 5.5 GHz) in Figure 1 are interchanged.

108. '... realises slow PS atoms...' This sounds like a conclusion before the results are presented.

110. I find the two diagrams in Fig 1b&c to be rather incomprehensible. There are 5 rectangles and only two labels 'iron shield' and 'magnetic lens'. The geometry of the chamber is completely mysterious. The scintillator is not mentioned in the main text. Many things are described in Methods, but anything appearing in the Figure should be described in the main article.

115. How many positrons per bunch? How consistent is the number? What is the repetition rate and bunch duration? These numbers are basic to the set-up and should be included. I see some details in Methods; but these should be listed here.

117. From the figure, it looks as if the PS used was emitted backwards towards the e+ beam. Is the number 3×10^3 for all directions?

123-137 I think it would be very helpful to have a diagram of the timings used here for positronium production, the various laser pulses used for cooling and probing, and the positron detection. What is the drift time of the positrons to the MCP, for example?

159. 20 minutes of measurement but we have no idea how many cycles this is - until reading Methods

167. What is the uncertainty on the quoted temperature of 600 K, and what contributes to it?

168. there is a reference to time-of-flight measurements but these are not described, nor is a reference given.

177. Error bars originate from... This statement is unhelpful. An accounting of uncertainties in the S variable is totally absent here; this needs to be remedied before publication can be considered seriously.

179. Figure 2a. spectral width of the laser: the figure shows a frequency resolution of about 100 GHz. Where does this come from? The laser was described as a 'nanosecond pulse' earlier.

181. Fig. 2b: The caption talks about velocity distribution, but the horizontal axis is frequency. According to the frequency resolution depicted in the panel, the points are oversampled by a factor of 2 or 3. Why is that?

191. How was 100 ns chosen for the cooling time? Likewise, the detuning range and chirp rate - how were they chosen?

196. 'narrowing the spectrum' This is a major component of the diagnostics here, and perhaps deserves

a mention in the main text as well as the description in Methods. It seems cryptic otherwise.

201. The analysis of the fractional change in the velocity distribution is crucially important here. It appears in Methods but there is no reference to the procedure here. It isn't clear from the main text if the higher resolution laser was used for both on and off data in Fig 2b, or if the off data come from a fit to Fig 2a. This should be made explicit in the main text or the Fig2b caption.

203. 'The optical frequency detuning...' Doesn't this belong in the previous paragraph?

237-287 The simulated results are not directly compared to the experimental data. Why is this? This section seems disjointed without a comparison of theory and data. The simulations are plotted versus velocity rather than laser frequency. At a minimum, having the same x-axis as the data would aid comparison. Why not convolve the calculated velocity distributions with the laser resolution to mimic a spectrum like Figure 2? I honestly don't understand the usefulness of the simulation as it stands.

289ff: Conclusions: The annihilation of positronium and its impact on the experiment is barely mentioned here. It would be useful to discuss the limits set by the finite lifetime when the cooling takes at least 100 ns. Such considerations are important for the future experiments alluded to here.

Referee #3 (Remarks to the Author):

This is a well-written paper describing an experiment which has achieved, for the first time, laser cooling in one dimension of a sample of positronium (Ps) atoms (the quasi-stable bound state comprised of an electron-positron pair). This is a very important advance which might well influence future studies of the atom, particularly those impacting on fundamental physics, such as spectroscopy. The work will be of interest beyond the field of low energy antimatter physics (typically to the cold atom/molecule community, which has a growing number of groups working on fundamental systems), and it is noted that Ps cooling has been a long-standing goal in that area, as explained in the introduction of the manuscript.

The paper is worthy of consideration for publication in Nature, pending satisfactory responses to the comments below.

Main text

Line 28. Reference 11 here is to Deutsch's original 1951 paper, which will not give the interested reader much information on Ps. Perhaps also include a recent review article, such as that in Eur. Phys. D by D.B. Cassidy.

Line 30. The fact that the Ps has been cooled in one dimension should be explicitly included here.

Line 54. The phrase "densified to a satisfactory level" is both awkward and meaningless. What is meant by densified and what will be satisfactory? This needs to be rewritten or deleted.

Line 62-3. The claim here is not strictly correct, since the antiprotons also have to be "cold" – and just to say "cold" is not useful in this context. Also, this reviewer does not believe a major aim of reference 38 was as stated in lines 64-5.

Line 117. The Ps are claimed to be "thermalised", though later in the paper their effective temperature is given as 600 K. Delete "thermalised".

Line 167. There should be an uncertainty on the quoted Ps temperature.

Line 177. Are the statistical and systematic uncertainties mentioned here discussed anywhere in the article? I recommend that a short section be added to the Methods where these are brought together for consideration.

Line 187. Is the "phenomenological model" discussed in the Methods? If so, reference that here; if not give some details.

Line 213. Here again we have a vague statement as "phenomenological function". Is this discussed in the

Methods? It looks as though it is, so please reference here.

Line 219. The phrase “approximately 1 K or sub-kelvin temperatures” seems to be contradictory. Please either amend or explain.

Lines 224-7. It would be useful here to have some kind of estimate of what fraction of the Ps emitted from the aerogel has been cooled. Why is the delayed fraction “unknown”. Surely this can be estimated with reasonable accuracy using timing methods at the disposal of this experiment.

Lines 258-60. Here the statement that the simulations shown in figure 3 agree “quantitatively” with the distribution shown in figure 2b needs to be amplified. Can a direct comparison be made so the level of agreement can be quantified. It should be possible if they agree “quantitatively”, as is claimed.

Methods Sections

This reviewer counted just 4 references to “Methods” in the main text, though there are 8 sections in the manuscript. They should all be referred to in the text as appropriate.

Line 546 and surrounding text. It is not clear from the text how P_e was used as claimed to set the functional form of the S parameter. Some more explanation is needed here.

Lines 550-1. Reference is made to Equations 1 and 2 here, but the equations are not numbered as far as this reviewer could see.

Line 571. Is the parameter “A” here the same as that used in the equation on line 538? One presumes so, but perhaps this can be mentioned.

Line 608. Extended Data fig 3 appears in this text before figure 2. Perhaps they can be reordered.

Lines 767-8. This figure caption is way too brief to be useful. Please expand to properly define parameters such as d_x , M_g etc. Even if they are defined elsewhere including them here will spare the reader from excessive cross-checking.

Lines 776-9. Replace “as identical” with “from” or something else suitable. Also, it is not clear what is meant by the “interval”. Is it the band? This sentence is very unclear to this reviewer, and needs clarifying.

Author Rebuttals to Initial Comments:

Note to all the referees:

We would like to express our sincere gratitude for the time and effort the referees have dedicated to reviewing our manuscript.

During the current review process, a paper on a similar subject was published in another journal, having been submitted on the same day as our manuscript. This explores one-dimensional laser cooling of positronium with a broadband laser pulse and reports a cooling result of approximately 170 K. In contrast, our innovative method using chirped pulse trains achieved a significantly colder 1 K. Our manuscript further evaluates the potential for sub-Doppler cooling based on our simulations.

Since results from both groups were presented at the same international conference, POSMOL 2023, on August 5, 2023 with the same submission date to journals, this demonstrates concurrent developments in Ps laser cooling. Therefore we have maintained in our revised manuscript the claim of being the first to demonstrate laser cooling of Ps, as well as that regarding the formation of Ps at unprecedented low velocities, as stated in our original manuscript.

In light of these facts, we have maintained in our revised manuscript the claim of being the first to demonstrate laser cooling of Ps, as well as the claim regarding the formation of Ps at unprecedented low velocities, as stated in our original manuscript.

As listed as numbers (72), (106) and (107) in the List of Changes, we have added a mention of that publication at the end of our manuscript.

Referee #1 (Remarks to the Author):

This is the first exhibition of the partial laser cooling of positronium. Could be useful for precision measurements of the 1S-2S interval of positronium, maybe not so useful for making a BEC.

We are profoundly grateful for the Referee's evaluation of our manuscript, especially on the usefulness for precision measurements.

We have comprehensively revised the manuscript, enhancing its readability, the quantitative nature of the discussion, and the depth of the arguments supported by numerical simulations, among other aspects. We trust that the referee will recognise the improved quality of the manuscript.

Referee #2 (Remarks to the Author):

This article reports one-dimensional laser cooling of positronium using an innovative frequency-chirped, pulsed laser. The authors demonstration of laser cooling involves exposing a pulsed positronium sample to the chirped laser for 100 ns. A probe setup consists of a pulsed, 243 nm laser to drive the 1S-2P transition, and a nanosecond-pulsed laser at 532 nm to ionise the 2P state. The positrons released in the ionisation are counted on a multichannel plate. The evidence for laser cooling is a single figure in which an uncooled spectrum is compared to a laser-cooled spectrum. The authors have examined only the central region of the Doppler profile for the cooled sample, and their claim is based upon increase of the signal over a narrow frequency range around zero detuning. A corresponding decrease in signal just outside of the enhanced region is presumably due to atoms that have been cooled towards the central region. The authors also present a simulation of the process in Figure 3.

This article represents a technical tour-de-force in the preparation and manipulation of positronium. The methods appear to be valid and original. However, I hesitate to recommend publication in Nature at this stage because of the small amount of evidence presented compared to the importance of the claim.

First and foremost, we express our profound gratitude to the Referee for the constructive critique of our manuscript demonstrating one-dimensional laser cooling of positronium.

In particular, the focus on only the central region of the velocity distribution in Figure 2b is concerning. It is clear that the effects of cooling should be visible in this region, but why not probe the whole profile, as in Figure 2a? This is, in my opinion, a serious oversight. Figure 2b shows a signal enhancement of a factor of more than two at zero relative frequency. This would be clearly visible on a broad scan, and the reader wouldn't be left wondering what happened to the rest of the distribution. A zoomed-in figure as shown could also be added.

Experiments probing the fractional change of the Doppler profile over an extended frequency range have not been performed. In the revised manuscript, we have made the following modifications to ensure that readers understand that the absence of the fractional change of the full profile is not an oversight but rather that we present data that is both necessary and sufficient to demonstrate laser cooling:

- **We have included a statement in the main text that the design of the cooling laser conditions for this experiment aimed to maximise the number of atoms around zero velocity. We have also noted that although it is feasible to target the entire Doppler distribution for cooling, it is not appropriate**

for the aim. (Please see (24), (44) and (45) in the List of Changes).

- To indicate the expected change in the entire Doppler profile, the frequency range displayed for the simulated Doppler profile after cooling has been extended and it has been compared with the Doppler profile without cooling (Extended Data Fig. 5 in the revised manuscript). Corresponding additions have been made to prompt references to the Methods. (Please see (57) and (104) of the List of Changes)
- We have made several modifications so that there is no ambiguity regarding the conditions of the laser cooling and control experiments. (Please see (46), (48), (55) and (81) of the List of Changes)
- The main text now includes a note that, with an increase in beam time and the number of positrons, it will become feasible to conduct experiments detecting the extremely low-temperature velocity distributions that are illustrated in our simulations. (Please see (63) of the List of Changes)

I am also missing a description of how such a two-component velocity distribution would be useful in future experiments. Surely the promise of laser cooling is fully realised when the whole distribution is cooled. The authors don't comment on this or outline how to achieve it in the future.

We have supplemented the main text with an explanation of the potential usefulness of a two-component velocity distribution for future experimental research. Additionally, we have noted in the main text that it is possible to achieve cooling of the entire velocity distribution, with an accompanying brief discussion of the strategy. To clarify the anticipated cooling results from adjustments to the cooling laser settings intended to cool the entire velocity distribution, we have added the results of simulations in Methods.

Please see (64), (67) and (105) in the List of Changes.

I am also missing - in the main text - some quantitative measure - with uncertainties - of the efficacy of the laser cooling. Cooled temperatures of 0.8 and 1.4K are quoted, but the nature of the uncertainties in this article is generally hard to find.

We have reorganised the logic of the relevant paragraph to give a more direct introduction of the efficacy of laser cooling. We have also added a description to incorporate the association of the evaluation of the experimental results with numerical simulations. Furthermore, we have included a sentence in the concluding paragraph that summarises the elements characterising the results of our

laser cooling experiments. We have added specific information regarding the nature of uncertainty in the experiments to the main text and Methods.

Please see (35), (53), (62) and (92) in the List of Changes.

While this work is exciting and innovative, I am not yet convinced that it is advanced enough to live up to the claim of the first laser cooling of positronium. We are also missing any characterisation of the degrees of freedom perpendicular to the cooling laser. It would also be good, but perhaps not absolutely necessary, to see another spectrum where the authors have varied something - say detuning or chirp rate - and seen an expected response.

We have added the Doppler profile in the direction normal to the surface of the silica aerogel to Extended Data Fig. 1. The experiment was done using a beam time after the initial submission. We have also added our interpretation of the profile and the reason for the absence of fitting to Methods, as the origin of the velocity distribution of this degree of freedom is not simple.

Although we are unable to add experiments with altered cooling laser conditions now, we have presented the expected velocity distributions after cooling using the framework of the simulations in Fig. 3. Specifically, we have added figures in the Methods showing simulated results for durations of cooling other than 100 ns, as well as for the case where the cooling duration was fixed at 100 ns and the chirp rate was varied. The agreement between the experimental results and simulations as presented in Fig. 3 guarantees that these are reliable predictions.

Please see (97), (101), and (105) in the List of Changes.

I have added some detailed comments below. They should be addressed in any future versions of the submission.

In the following, I refer to the line numbers in the Nature-provided PDF file.

We greatly appreciate the Referee's constructive and detailed comments. We have carefully considered all the comments and reflected these considerations in the revised manuscript.

25. What does 'direction independent' mean? Please revise.

We have replaced “direction independent spontaneous emission” with “spontaneous emission in random directions”.

Please see (2) in the List of Changes.

39. It is not clear that progress in this field has anything to do with matter-antimatter asymmetry. This is a motivation, but there is as yet no obvious link.

We concur with the Referee's comment and have removed the sentence. Refs. 13 and 14 are retained in the revised manuscript in the section discussing matter-antimatter asymmetry following the abstract.

Please see (4) in the List of Changes.

43. reference needed for BEC of positronium

We have added the references 27-31 cited in Line 56 of the previous manuscript to the corresponding sentence.

Please see (6) in the List of Changes.

48-49. Be quantitative about the current precision for experiment and theory.

In the revised manuscript, we have included both experimental and theoretical figures for the relative uncertainty of the 1S-2S transition frequency, namely 2.6×10^{-9} and 4.7×10^{-10} , respectively. Additionally, we found it appropriate to cite Ref. 45 in the previous manuscript in relation to the theoretical calculations and have therefore added this citation here.

Please see (7) in the List of Changes.

54. What about antiprotonic helium? What does 'a satisfactory level' mean

The statement was made in relation to the critical density of the BEC; with consideration of the question from the Referee and the recommendation from Referee #3, we have deleted this part.

Please see (8) in the List of Changes.

56. What does high temperature mean here? You should be very explicit for the Nature readership.

The discussions of previous studies ranged from a few kelvins to several tens of kelvins - we have added this to the text.

Please see (9) in the List of Changes.

66-73. This description of laser cooling is not very illuminating, and it applies only to systems at rest in the lab. There is no mention of the Doppler effect and laser detuning here. Zeeman slowing is not cooling.

We have modified the corresponding part to explain that the laser cooling is based on photon recoil and the Doppler effect. We also described that moving particles absorb oppositely propagating red-detuned laser photons. We removed the Zeeman slower from the example list of laser cooling methods.

Please see (11) and (12) in the List of Changes.

80. It would be helpful to make a temperature/velocity comparison here for poistronium. E.G. what is the recoil velocity in equivalent temperature units.

We have added “(equivalent to 55 mK)” to the corresponding sentence.

Please see (14) in the List of Changes.

82. You should explicitly state the natural linewidth here. this is key to all of the future discussion.

We have modified “the natural linewidth” to “the natural linewidth of 50 MHz”.

Please see (16) in the List of Changes.

83. I think you mean 'narrow bandwidth' and not single frequency here.

We have replaced “single-frequency” with “narrow-bandwidth”.

Please see (17) in the List of Changes.

89. Is there a reference needed here for the 10K limit?

We have added a citation (Ref. 38 in the previous manuscript) to the sentence in question, as per the Referee’s suggestion.

Please see (18) in the List of Changes.

98. There is a reference to a 'tailored laser (see Fig 1a)', but there is nothing about the laser in the figure except the pulse spectra. I was expecting a depiction of the laser system.

We have added information to the corresponding sentence that concisely describes the operating principle of our cooling laser, encouraging reference to Methods. Additionally, we have added a new section in Method that briefly introduces the characteristics of the cooling laser.

Please see (20) and (88) in the List of Changes.

98. 'short optical pulses' is not helpful; be quantitative

We have incorporated "approximately 0.1 ns duration" into the corresponding sentence. This is related to our response to the Referee's previous comment.

Please see (20) in the List of Changes.

100. The frequency widths between the three 2P levels (4.4 and 5.5 GHz) in Figure 1 are interchanged.

As the Referee pointed out, the two numbers were interchanged. We have modified Fig. 1a.

Please see (73) in the List of Changes.

108. '... realises slow PS atoms...' This sounds like a conclusion before the results are presented.

We have replaced "realises" with "is anticipated to produce".

Please see (26) in the List of Changes.

110. I find the two diagrams in Fig 1b&c to be rather incomprehensible. There are 5 rectangles and only two labels 'iron shield' and 'magnetic lens'. The geometry of the chamber is completely mysterious. The scintillator is not mentioned in the main text. Many things are described in Methods, but anything appearing in the Figure should be described in the main article.

We have modified Figure 1b to indicate the position of the vacuum chamber walls clearly and to render the magnetic lens in a comprehensive manner. We have mentioned the magnetic lens, the scintillators and their applications in the main text.

Please see (27), (28) and (74) in the List of Changes.

115. How many positrons per bunch? How consistent is the number? What is the repetition rate and bunch duration? These numbers are basic to the set-up and should be included. I see some details in Methods; but these should be listed here.

In the revised manuscript, we have added “(16 ns duration, 50 Hz repetition rate)” to the corresponding sentence. We have added to Methods a sentence describing that the instability in the number of positrons remained at approximately 1% throughout the experiment.

Please see (27) and (87) in the List of Changes.

117. From the figure, it looks as if the PS used was emitted backwards towards the e+ beam. Is the number 3×10^3 for all directions?

As noted by the Referee, Ps generated in the direction opposite to the direction of flight of the e+ bunch was used, with 3×10^3 indicating the number of Ps emitted in that direction. Almost all Ps emitted from the silica aerogel are in that direction, and very little is emitted from the opposite side of the silica aerogel.

To more accurately convey the experimental arrangement, we have modified the corresponding sentence.

Please see (30) in the List of Changes.

123-137 I think it would be very helpful to have a diagram of the timings used here for positronium production, the various laser pulses used for cooling and probing, and the positron detection. What is the drift time of the positrons to the MCP, for example?

Following the Referee's recommendation, we have added a timing chart as Fig. 2c. The Fig. 2c in the previous manuscript has been incorporated as an inset into Fig. 2b. We have also added detailed information on the drift time to Methods.

Please see (34), (75) and (91) in the List of Changes.

159. 20 minutes of measurement but we have no idea how many cycles this is - until reading Methods

We have added to the main text the number of measurement cycles corresponding to a 20-minute measurement period. Additionally, we have newly incorporated into the main text that in the laser cooling experiment, a measurement time of 4 hours was required for each frequency component of the fractional change, and information regarding how many cooling events this equates to has been added. Furthermore, we have included references to the Methods section in the corresponding sentences.

Please see (38), (48) and (93) in the List of Changes.

167. What is the uncertainty on the quoted temperature of 600 K, and what contributes to it?

The uncertainty in the evaluated temperature is 50 K. We have amended the number of the temperature to $6.1(5)\times 10^2$ K. We also corrected an error found in our rounding procedure, replacing 6.0 with 6.1. Furthermore, this uncertainty is attributed to the statistical uncertainty in the number of ionised positrons, which is a measure of the signal in our Doppler spectroscopy measurements. We have added this explanation to the main text.

Please see (35) and (42) in the List of Changes.

168. there is a reference to time-of-flight measurements but these are not described, nor is a reference given.

Regarding the mention of time-of-flight measurements, we currently lack suitable references for citation, which complicates providing an appropriate explanation. Consequently, we have substituted the specific mention with a more generalised description accompanied by a citation. This alteration does not impact the discussion in our manuscript; rather, we are convinced that it has resulted in a beneficial modification, enriching the physical insight more than before.

Please see (43) in the List of Changes.

177. Error bars originate from... This statement is unhelpful. An accounting of uncertainties in the S variable is totally absent here; this needs to be remedied before publication can be considered seriously.

The uncertainties in the experimental data we presented are dominated by statistical uncertainties, originating from the statistical randomness in the number of ionised positrons observed in the measurement of Doppler profiles and their fractional changes.

We have modified the figure caption of Fig. 2a accordingly. The contribution of the systematic uncertainty in the present experiment is negligible compared to the statistical uncertainty, hence the description of the systematic uncertainty has been omitted. Additionally, we have included a detailed description of the origin of the statistical uncertainties in the final part of the “Doppler spectroscopy” section in Methods.

Please see (35), (79) and (92) in the List of Changes.

179. Figure 2a. spectral width of the laser: the figure shows a frequency resolution of about 100 GHz. Where does this come from? The laser was described as a 'nanosecond pulse' earlier.

We have revised the main text and Methods to clarify that the spectral width of the probe pulse originates from the longitudinally multimode nature.

Please see (41), (77) and (90) in the List of Changes.

181. Fig. 2b: The caption talks about velocity distribution, but the horizontal axis is frequency. According to the frequency resolution depicted in the panel, the points are oversampled by a factor of 2 or 3. Why is that?

First, in the section of the manuscript where Fig. 2b is introduced and in its caption, we have replaced the term "the velocity distribution" with "the Doppler profile".

The frequency resolution displayed in Fig. 2b of the previous manuscript was unclear as to whether it was the one used in the typical fitting. In the revised manuscript, we have shown in Fig. 2b both the range of frequency resolution employed in the experiments and the frequency resolution used for fitting. The number of measurement points was appropriate for acquiring spectral structures limited by frequency resolution, if we assume that all the data points could be acquired with the best resolution of 8 GHz. Therefore, the experimental setting did not constitute oversampling.

Please see (46), (49), (77), (78) and (80) in the List of Changes.

191. How was 100 ns chosen for the cooling time? Likewise, the detuning range and chirp rate - how were they chosen?

In the main text, we have added a brief description of our approach to maximising the number of atoms near zero velocity and encouraged readers to consult the Methods section for more details. In the Methods section, we have included an explanation and added Extended Data Fig. 6 to explicitly show that the 100 ns duration and the approximately 500 GHz/ μ s chirp rate of the cooling laser were chosen to maximise the number of atoms near zero velocity through cooling.

Please see (44) and (105) in the List of Changes.

196. 'narrowing the spectrum' This is a major component of the diagnostics here, and perhaps deserves a mention in the main text as well as the description in Methods. It seems cryptic otherwise.

We have added information to the sentence to clarify that we achieved spectral narrowing using a Fabry–Pérot etalon.

Please see (47) in the List of Changes.

201. The analysis of the fractional change in the velocity distribution is crucially important here. It appears in Methods but there is no reference to the procedure here. It isn't clear from the main text if the higher resolution laser was used for both on and off data in Fig 2b, or if the off data come from a fit to Fig 2a. This should be made explicit in the main text or the Fig2b caption.

To address the Referee's comments, we have updated the paragraph in the main text that introduces the analysis of the fractional change by adding "(see Methods)". A sentence has been appended to the caption of Fig. 2b to convey that the probe pulse with a narrowed linewidth was employed in both the cooling and control experiments.

Please see (51), (82) and (83) in the List of Changes.

203. 'The optical frequency detuning...' Doesn't this belong in the previous paragraph?

The sentence the Referee pointed out describes how the optical frequency of the cooling laser changes, and it was appropriately placed in the relevant paragraph. The preceding paragraph details the frequency range and resolution at which changes in the Doppler profile were observed.

To ensure that our intent is clearly conveyed to readers, we have added "of the cooling laser" to the sentence.

Please see (50) in the List of Changes.

237-287 The simulated results are not directly compared to the experimental data. Why is this? This section seems disjointed without a comparison of theory and data. The simulations are plotted versus velocity rather than laser frequency. At a minimum, having the same x-axis as the data would aid comparison. Why not convolve the calculated velocity distributions with the laser resolution to mimic a spectrum like Figure 2? I honestly don't understand the usefulness of the simulation as it stands.

We have moved Extended Data Fig. 4 in the previous manuscript to the main text, as Fig. 3b. The figure enables direct comparison with the experimental data.

To strengthen the usefulness of the simulation, we have added references and a brief description on the possibility of achieving sub-recoil cooling. This could be of significant interest for future precision spectroscopy studies of Ps.

We have also added to the main text the reason for presenting Fig. 3 (Fig. 3a in the revised manuscript) as a velocity distribution rather than a Doppler profile. The velocity representation can help readers estimate the transit-time broadening and second-order Doppler shifts in precision spectroscopy.

Please see (57), (61), (85) and (106) in the List of Changes.

289ff: Conclusions: The annihilation of positronium and its impact on the experiment is barely mentioned here. It would be useful to discuss the limits set by the finite lifetime when the cooling takes at least 100 ns. Such considerations are important for the future experiments alluded to here.

The potential impact of the 142 ns lifetime of Ps on our chirp cooling is related to the Referee's comment concerning the cooling of the entire velocity distribution, and a brief discussion has been added to the concluding paragraph of the main text. To convey to readers the velocity distributions resulting from the influence of lifetime with extended cooling times, specific simulation results have been additionally introduced in Methods. A reference to Methods has also been added.

Please see (64) and (105) in the List of Changes.

Referee #3 (Remarks to the Author):

This is a well-written paper describing an experiment which has achieved, for the first time, laser cooling in one dimension of a sample of positronium (Ps) atoms (the quasi-stable bound state comprised of an electron-positron pair). This is a very important advance which might well influence future studies of the atom, particularly those impacting on fundamental physics, such as spectroscopy. The work will be of interest beyond the field of low energy antimatter physics (typically to the cold atom/molecule community, which has a growing number of groups working on fundamental systems), and it is noted that Ps cooling has been a long-standing goal in that area, as explained in the introduction of the manuscript.

The paper is worthy of consideration for publication in Nature, pending satisfactory responses to the comments below.

First and foremost, we would like to express our deepest gratitude for the Referee's substantial dedication to reviewing our manuscript.

We have carefully considered all the comments the Referee gave us and have incorporated them into our revised manuscript. Below, we present our responses to each comment and detail the amendments made to the manuscript.

Main text

Line 28. Reference 11 here is to Deutsch's original 1951 paper, which will not give the interested reader much information on Ps. Perhaps also include a recent review article, such as that in Eur. Phys. D by D.B. Cassidy.

We have added the suggested reference from the Referee to the corresponding sentence as recommended.

Please see (106) in the List of Changes.

Line 30. The fact that the Ps has been cooled in one dimension should be explicitly included here.

We have added “one-dimensional” to the corresponding sentence.

Please see (3) in the List of Changes.

Line 54. The phrase “densified to a satisfactory level” is both awkward and meaningless. What is meant by densified and what will be satisfactory? This needs to be rewritten or deleted.

We have removed the sentence.

Please see (8) in the List of Changes.

Line 62-3. The claim here is not strictly correct, since the antiprotons also have to be “cold” – and just to say “cold” is not useful in this context. Also, this reviewer does not believe a major aim of reference 38 was as stated in lines 64-5.

We have modified the sentence. Additionally, in the sentence where we cited Ref. 38 in the previous manuscript, we have removed the phrase “for that purpose.”

Please see (10) in the List of Changes.

Line 117. The Ps are claimed to be “thermalised”, though later in the paper their effective temperature is given as 600 K. Delete “thermalised”.

We have removed “thermalised” from the sentence. We have also added a description of the origin of the temperature difference between the Ps gas and the silica aerogel.

Please see (29) and (43) in the List of Changes.

Line 167. There should be an uncertainty on the quoted Ps temperature.

The uncertainty in the evaluated gas temperature is 50 K. In the revised manuscript, the gas temperature is now $6.1(5)\times 10^2$ K. At the same time, we have corrected an error found in our rounding procedure, replacing 6.0 with 6.1.

We have also added sentences to the main text and Methods explaining that the dominating source of the temperature uncertainty is the statistical uncertainty in the number of detected positrons. This is related to our response to the Referee’s next comment.

Please see (35), (42) and (92) in the List of Changes.

Line 177. Are the statistical and systematic uncertainties mentioned here discussed anywhere in the article? I recommend that a short section be added to the Methods where these are brought together for consideration.

In the case of our experiment, the measurement uncertainty is dominated entirely by statistical uncertainty. Therefore, we have chosen to limit our description to statistical uncertainties only, with their origins detailed in the main text and the Methods.

Please see (35) and (92) in the List of Changes.

Line 187. Is the “phenomenological model” discussed in the Methods? If so, reference that here; if not give some details.

Regarding the "phenomenological model," we had discussed it in Methods. However, the title of the relevant section was "Analysis of the fractional change in the Doppler profile," which did not directly refer to the model. We have now added "(see Methods)" in the figure caption and modified the title.

Please see (82) and (98) in the List of Changes.

Line 213. Here again we have a vague statement as “phenomenological function”. Is this discussed in the Methods? It looks as though it is, so please reference here.

We have added “(see Methods)” to the relevant sentence and have replaced “function” with “model” for consistency.

Please see (51) in the List of Changes.

Line 219. The phrase “approximately 1 K or sub-kelvin temperatures” seems to be contradictory. Please either amend or explain.

We agree with the Referee’s opinion and have removed “or sub-kelvin temperatures”.

Please see (52) in the List of Changes.

Lines 224-7. It would be useful here to have some kind of estimate of what fraction of the Ps emitted from the aerogel has been cooled. Why is the delayed fraction “unknown”. Surely this can be estimated with reasonable accuracy using timing methods at the disposal of this experiment.

We do not possess additional data on this matter, and we are not able to conduct further experiments at present due to beam time scheduling. However, using simulations shown in Fig. 3a that assume no release of delayed Ps, we have quantified the reduction in the proportion of the region resonant with the cooling laser compared to the non-cooled case. We have also determined what percentage of the total population this reduction corresponds to, as detailed in the main text. Additionally, we have added a few sentences in the Methods section. These describe our empirical observations, which suggest the presence of Ps emitted from the silica aerogel several tens of nanoseconds after the injection of the positron bunch. We observed the components of zero lateral velocity, although the observation is qualitative.

Please see (54), (99) and (109) in the List of Changes.

Lines 258-60. Here the statement that the simulations shown in figure 3 agree “quantitatively” with the distribution shown in figure 2b needs to be amplified. Can a direct comparison be made so the level of agreement can be quantified. It should be possible if they agree “quantitatively”, as is claimed.

The Extended Data Fig. 4 in the previous manuscript indeed showed the quantitative agreement between the experimental results and numerical simulations. Therefore, we have decided to employ this figure as Fig. 3b in the revised manuscript.

Please see (85) in the List of Changes.

Methods Sections

This reviewer counted just 4 references to “Methods” in the main text, though there are 8 sections in the manuscript. They should all be referred to in the text as appropriate.

We have added references in the main text to all the sections in the Methods.

Please see (22), (28), (38), (41), (44), (47), (48), (51), (54), (64) and (82) in the List of Changes.

Line 546 and surrounding text. It is not clear from the text how P_e was used as claimed to set the functional form of the S parameter. Some more explanation is needed here.

P_e is the equation we referred to when constructing the functional form of the S parameter. P_e was not utilised in the S parameter itself. To clarify how we referenced P_e , we have added the expression “(see the denominator in the fraction)” to the corresponding sentence.

Please see (95) in the List of Changes.

Lines 550-1. Reference is made to Equations 1 and 2 here, but the equations are not numbered as far as this reviewer could see.

We have replaced “Eq. (1)” with “ $S(\omega_R)$ ”, and “Eq. (2)” with “ P_e ”.

Please see (96) in the List of Changes.

Line 571. Is the parameter "A" here the same as that used in the equation on line 538? One presumes so, but perhaps this can be mentioned.

In the equation at line 571 of the previous manuscript, the parameter "A" was used to characterise the signal magnitude in Doppler profile measurements at a frequency resolution of 110 GHz, evaluated without employing laser cooling to measure the gas temperature. In contrast, the "A" used in the equation at line 538 referred to a coefficient characterising the signal magnitude of components that were cooled via laser cooling. Consequently, the two parameters are distinct. In the revised manuscript, we have substituted the latter coefficient of "A" with "C" to differentiate between the two clearly.

Please see (94) in the List of Changes.

Line 608. Extended Data fig 3 appears in this text before figure 2. Perhaps they can be reordered.

Indeed, the sequence of citations for the two figures had been inadvertently reversed. In the revised manuscript, we have addressed this by swapping the two Figures and correcting their citations accordingly.

Please see (102) in the List of Changes.

Lines 767-8. This figure caption is way too brief to be useful. Please expand to properly define parameters such as d_x , M_g etc. Even if they are defined elsewhere including them here will spare the reader from excessive cross-checking.

We have revised to ensure that the figure caption accurately describes the definitions of each symbol and the meaning of the numbers depicted within the figures.

Please see (103) in the List of Changes.

Lines 776-9. Replace “as identical” with “from” or something else suitable. Also, it is not clear what is meant by the “interval”. Is it the band? This sentence is very unclear to this reviewer, and needs clarifying.

As we have previously mentioned, the Extended Data Fig. 4 of the previous manuscript has been adopted as Figure 3b in the revised manuscript.

We have replaced “as identical” with “from” as the Referee pointed out. Also, we have revised the figure captions to clarify that terms such as "interval" and "filled band" represent the thickness of the curve.

Please see (85) and (100) in the List of Changes.

List of Changes

Main text:

- (1) We have edited the abstract from the perspectives of readability and word count reduction.
- (2) We have replaced “direction independent spontaneous emission” in the abstract with “spontaneous emission in random directions”. (line 25 of the previous manuscript)
- (3) We have replaced “Here, we demonstrate laser cooling of positronium.” in the abstract with “Here, we demonstrate one-dimensional laser cooling of positronium.” (line 30 of the previous manuscript)
- (4) We have removed the sentence “Progress in this field is vital in the search for the matter-antimatter asymmetry in the universe.” in the abstract (line 39 of the previous manuscript).
- (5) We have replaced “One-dimensional chirp cooling of the dilute positronium gas in a counter-propagating configuration gave a final velocity distribution corresponding to approximately 1 K in a short time of 100 ns.” in the abstract with “One-dimensional chirp cooling was used to cool a portion of the dilute positronium gas to a velocity distribution of approximately 1 K in 100 ns.” (line 34 of the previous manuscript)
- (6) We have cited references 27–31 of the previous manuscript in the sentence mentioning Bose–Einstein condensation in the abstract. (line 43 of the previous manuscript)
- (7) We have added fractional uncertainties of the experimental and theoretical 1S-2S transition frequency, i.e., 2.6×10^{-9} and 4.7×10^{-10} , to the subsequent paragraph of the abstract. In addition, we have cited Ref. 45 in the previous manuscript here. (line 49 of the previous manuscript)
- (8) We have removed the sentence “Currently, Ps is the only antiparticle-containing atomic system that can be densified to a satisfactory level in a gas phase.” in the subsequent paragraph of the abstract. (line 53 of the previous manuscript)
- (9) We have replaced the sentence “Owing to its light mass, only twice that of an electron, Bose-Einstein condensation (BEC) is expected to occur at relatively high temperatures²⁷⁻³¹ compared to ordinary atoms.” in the subsequent paragraph of the abstract with “Owing to its light mass, only twice that of an electron, Bose-Einstein condensation (BEC) is expected to occur at relatively high temperatures, ranging from a few kelvin to several tens of kelvin¹⁴⁻¹⁸,” (line 54 of the previous manuscript)
- (10) We have replaced the sentence “Cold Ps is also requisite for generating cold antihydrogen pulses, and laser cooling with a wide spectral width and long laser duration in a high magnetic field has recently been investigated theoretically for that purpose.” in the second paragraph after the abstract with “Furthermore, cold Ps can be used for efficient antihydrogen generation³⁶.” and “Laser cooling with a wide spectral width and long laser duration in a high magnetic field has recently been investigated theoretically⁴⁰.” The former has been moved to the last part of the preceding paragraph. (line 62 of the previous manuscript)
- (11) We have added “based on photon recoil and the Doppler effect.” and “red-detuned” to the paragraph introducing laser cooling. (line 66 of the previous manuscript)
- (12) We have removed “and Zeeman slower” and revised the corresponding sentence in the paragraph introducing laser cooling as “Examples of current laser cooling techniques include Doppler cooling, chirp cooling and magneto-optical traps.” (line 73 of the previous manuscript)
- (13) We have replaced “and the decay becomes faster in a magnetic field” with “, requiring rapid cooling” in the paragraph describing the difficulties in laser cooling of Ps. (line 76 of the previous manuscript)
- (14) We have added “equivalent to 55 mK” to the sentence in which we defined the recoil velocity of positronium. (line 80 of the previous manuscript)
- (15) We have added “The large recoil velocity enables rapid deceleration. However,” to the paragraph describing the difficulties in laser cooling of Ps. (line 80 of the previous manuscript)

- (16) We have added “of 50 MHz” to the sentence in which we compared the resonance frequency change of 6.2 GHz with the natural linewidth. (line 82 of the previous manuscript)
- (17) We have replaced “single-frequency” with “narrow-bandwidth” in the sentence where we made a comparison between the laser cooling of ordinary atoms and that of positronium. (line 83 of the previous manuscript)
- (18) We have cited Ref. 38 in the previous manuscript in the context of introducing the limitation of cooling to several tens of Kelvin when using a broadband laser with a linewidth of the order of 100 GHz. We have also removed “by the linewidth” from the sentence. (line 89 of the previous manuscript)
- (19) We have added “to overcome these challenges,” to the paragraph introducing our strategy for laser cooling of Ps. (line 90 of the previous manuscript)
- (20) We have replaced the first two sentences of the paragraph introducing the cooling laser with the following: “We used a previously demonstrated laser^{43,44} that has the potential to realise such cooling. From this tailored laser, Using this tailored laser, which is based on an injection-locked pulsed laser incorporating an electro-optic modulator in the laser cavity, short optical pulses of approximately 0.1 ns duration were successively output every 4.2 ns (see Fig. 1a and Methods).” (line 98 of the previous manuscript)
- (21) We have replaced “split over 9.87 GHz” with “triply split over 9.87 GHz” in the sentence describing the splitting the 1S–2P transition frequency. (line 100 of the previous manuscript)
- (22) We have added “(see Methods)” to the sentence describing the need for the repump. (line 101 of the previous manuscript)
- (23) We have replaced “varies” with “increases” in the sentence describing the frequency chirp of the cooling laser. (line 103 of the previous manuscript)
- (24) We have added “The duration of the pulse train was adjustable up to approximately 1 μ s.” to the paragraph introducing the cooling laser. (line 104 of the previous manuscript)
- (25) We have replaced the sentence introducing our cooling strategy for Ps with “By effecting progressive deceleration from fast to slow Ps with chirp cooling in a counter-propagating configuration, we expect to obtain a velocity distribution close to the recoil-limited one.” (line 107 of the previous manuscript)
- (26) We have replaced “This laser cooling method realises slow Ps atoms in...” in the sentence describing the characteristics of our cooling with “This laser cooling method is anticipated to produce slow Ps atoms in...” (line 108 of the previous manuscript).
- (27) We have replaced “Bunches of positrons were injected at room temperature into a silica aerogel, which was used as a Ps formation medium” with “Bunches of 10^4 positrons (16 ns duration, 50 Hz repetition rate) were guided through a magnetic lens and injected at room temperature into a silica aerogel, which was used as the Ps formation medium.” (line 115 of the previous manuscript)
- (28) The following sentence has been added to the section introducing the positron bunch: “Gamma-ray detection using scintillators facilitated the measurement of the arrival time of the positron bunch (see Methods).” (line 117 of the previous manuscript)
- (29) We have removed “thermalised” from the sentence in which we described the release of approximately 3000 positronium atoms per positron bunch from the silica aerogel (line 117 of the previous manuscript).
- (30) We have added “on the positron-injection side” to the sentence in which we described the release of approximately 3000 positronium atoms per positron bunch from the silica aerogel. (line 117 of the previous manuscript)
- (31) We have replaced “We introduced three collimated laser beams into the vacuum chamber, expanded to encompass the entirety of the spatial spread of Ps.” in the paragraph describing the experimental setup in the vacuum chamber with “The three collimated laser beams encompassed the entirety of

- the Ps spatial spread.” (line 121 of the previous manuscript)
- (32) We have replaced “The three laser beams consisted of a cooling laser pulse train at 243 nm, a nanosecond laser pulse at 243 nm for velocity distribution measurements via the 1S–2P transition, and a nanosecond laser pulse at 532 nm to ionise the 2P state” in the paragraph describing the signal detection in the Doppler profile measurements with “The laser beams consisted of the cooling laser at 243 nm, a nanosecond laser pulse at 243 nm, and a nanosecond laser pulse at 532 nm.” (line 123 of the previous manuscript)
 - (33) We have replaced “allowing measurements within a limited experimental beam time” with “enabling measurements to be conducted within a few days of beam time” in the sentence introducing the benefits of utilising an MCP. (line 133 of the previous manuscript)
 - (34) We have added “Fig. 1c is the timing chart from the positron injection for Ps generation to the acquisition of the voltage signal from the MCP.” to the paragraph introducing the positron detection using a microchannel plate. (line 135 of the previous manuscript)
 - (35) We have added “In the following, the uncertainty of the signal level is dominated by the statistical Poisson uncertainty associated with the detected number of photoionised positrons.” to the paragraph introducing the positron detection using a microchannel plate. (line 137 of the previous manuscript)
 - (36) We have removed the second sentence, “To evaluate temperature, we measured the Doppler profile in the 1S–2P excitation spectrum that reflects the velocity distribution due to the Doppler effect.” from the paragraph describing the temperature measurement. (line 154 of the previous manuscript)
 - (37) We have removed the following sentence from the paragraph describing the temperature evaluation using Doppler spectroscopy: “The Doppler profile was obtained by evaluating the excitation signal originating from the ionised positrons while varying the optical frequency of the pulsed laser that induced the 1S–2P transition (see Methods).” Correspondingly, we have made minor adjustments to the first sentence of the paragraph. (line 156 of the previous manuscript)
 - (38) We have added “(equivalent to 1.2×10^4 measurement cycles, see Methods)” to the sentence where we mentioned the measurement time of 20 minutes in the Doppler spectroscopy measurements. (line 159 of the previous manuscript)
 - (39) We have removed “The measured Doppler profile is shown in Fig. 2a.” from the paragraph describing the gas temperature measurements. (line 159 of the previous manuscript)
 - (40) We have simplified the sentence that describes our use of the Maxwell-Boltzmann distribution function to estimate the gas temperature to: “The gas temperature was estimated by assuming the Maxwell–Boltzmann distribution function.” (line 162 of the previous manuscript)
 - (41) We have replaced “Fitting the data with a model that accounts for the spectral width of the 1S-2P transition-inducing laser pulse and Lamb dip” describing the fitting in Fig. 2a as “Fitting the data with a model (see Methods) that accounts for a 110-GHz the spectral width of the multimode probe pulse and Lamb dip”. (line 164 in the previous manuscript).
 - (42) We have replaced “ 6.0×10^2 K” with “ $6.1(5) \times 10^2$ K”, and added “(the source of the uncertainty is statistical)” to the sentence we presented the measured temperature of the Ps gas. (line 167 in the previous manuscript)
 - (43) The sentence beginning with “The Ps atoms in the aerogel were released into vacuum before thermalisation was complete, which was also observed in our time-of-flight measurements” in the paragraph introducing the temperature measurement of the Ps gas has been replaced with the following sentences: “In aerogels with mean free paths of tens of nm⁴⁶, thermalisation is relatively slow. If Ps generated by low-energy positrons are emitted into vacuum before fully thermalising, this could account for the temperature difference between the aerogel and the Ps gas.” (line 167 in the previous manuscript)
 - (44) We have replaced “To demonstrate laser cooling, we irradiated the cooling laser for 100 ns, starting

with the peak time of the positron pulse injection into the aerogel.” with “To demonstrate laser cooling, we irradiated the Ps gas with the cooling laser for 100 ns with the aim of maximising the number of atoms around zero velocity (see Methods), commencing at the peak time of the positron pulse injection.” (line 191 of the previous manuscript)

- (45) We have added the following sentence to the paragraph describing the experimental conditions to demonstrate laser cooling: “Although it is possible to target the entire Doppler profile by extending the duration of the cooling laser, this is expected to yield fewer zero-velocity components (see Methods).” (line 194 of the previous manuscript)
- (46) We have added “to best investigate the changes in the components interacting with the cooling laser” to the sentence where we described the probed frequency range in the cooling experiments with an improved frequency resolution (line 194 of the previous manuscript).
- (47) In the sentence describing the spectral narrowing of the probe pulse, we have replaced “by narrowing the spectrum of the broadband laser pulse of a linewidth of 110 GHz (see Methods)” with “by narrowing the 110-GHz linewidth of the original probe pulse using a Fabry–Pérot etalon (see Methods)” (line 196 of the previous manuscript).
- (48) We have added “In the following, the measurement time for each frequency was 4 hours, which is equivalent to 7.2×10^4 measurement cycles (see Methods).” to the last part of the paragraph where we explained the experimental conditions for the cooling experiment. (line 197 in the previous manuscript)
- (49) We have removed the first sentence of the paragraph introducing the experimental results for laser cooling, “We measured the change in the Ps velocity distribution resulting from irradiation with the cooling laser.” (line 198 of the previous manuscript)
- (50) We have added “of the cooling laser” to the sentence in which we described the optical frequency of the cooling laser. (line 203 of the previous manuscript)
- (51) We have replaced “phenomenological function” with “phenomenological model (see Methods)” in the sentence where we described our fitting in the main text. We have also simplified the corresponding sentence. (line 213 of the previous manuscript)
- (52) We have removed “or sub-kelvin temperatures” from the sentence describing a conservative estimate of the temperature of the cooled components (line 219 of the previous manuscript).
- (53) We have reordered and modified the latter half of the paragraph describing the evaluation of the fractional change induced by the laser cooling. These amendments were made to more specifically highlight the efficacy of our laser cooling technique. (line 220 of the previous manuscript)
- (54) We have replaced the sentence beginning with “Although the unknown number of delayed Ps released from the silica aerogel” with the following sentence: “Although the unknown number of delayed Ps released from the silica aerogel (see Methods) precludes further quantitative discussion, most of the Ps resonating with the cooling laser were efficiently decelerated, as supported by the subsequent simulation (approximately 70% reduction in the swept frequency region, resulting in approximately 10% of the entire Ps population constituting the cooled component, assuming no delayed Ps release).” (line 225 of the previous manuscript)
- (55) We have added “and no fractional change was expected” to “In this case, the Ps in the probed velocity range did not resonate with the cooling laser.” in the paragraph where we described the control experiment. (line 230 of the previous manuscript)
- (56) We have replaced the last sentence of the paragraph introducing the control experiment with the following: “Thus, the cooling laser does not impart a velocity change to the Ps that are off-resonant with it.” (line 234 of the previous manuscript)
- (57) We have replaced the first two sentences introducing the numerical simulation with the following: “In Doppler spectroscopy, velocity distribution assessment is influenced by frequency resolution and 2P level splitting. To examine the presumed velocity distribution indicated in the experimental

- results, we conducted numerical simulations. We developed a framework to calculate the time evolution of the density matrix based on the Lindblad master equation (see Methods).” (line 237 of the previous manuscript)
- (58) We have replaced “because of the dilute nature of the Ps gas” in the paragraph introducing the framework of our numerical simulation with “and delayed Ps release from the aerogel” (line 243 of the previous manuscript)
 - (59) We have removed the sentence “We plotted the probability density for each velocity with respect to that at zero at the beginning of the pulse train.” from the paragraph introducing the simulated results shown in Fig. 3a. The information equivalent to the deleted sentence is described in the figure caption of Fig. 3a. (line 248 of the previous manuscript)
 - (60) We have modified the second sentence in the paragraph describing the numerical simulation after the end of the cooling laser as “Considering the frequency resolution and the possible delayed Ps release inferred from the fitting above, the simulated distribution quantitatively reproduced (shown in Fig. 3b) the fractional change in the Doppler profile shown in Fig. 2b.” (line 258 of the previous manuscript)
 - (61) We have replaced the latter half of the paragraph describing the discrete peaks in the simulated velocity distribution with “Although further study is needed, these structures indicate a sub-recoil cooling capability, similar to the dark states in Doppler cooling^{47,48}. State-selective cooling in the final stage will be beneficial for the realisation of a sub-100 mK Ps gas, eliminating the shifted peaks and transitions that induce residual acceleration.” (line 268 of the previous manuscript)
 - (62) We have replaced the second sentence in the concluding paragraph with “This novel chirped cooling method has been proven to efficiently cool Ps to a hitherto unexplored narrow velocity range near zero, thus opening the scientific opportunities of this low-temperature, fundamental matter-antimatter bound system.” (line 290 in the previous manuscript)
 - (63) We have added “Increased positron number and longer beam time will enable verification of the sub-Doppler, narrow components indicated by simulations.” to the concluding paragraph of the main text. (line 292 of the previous manuscript)
 - (64) We have added “Several strategies can be taken to cool the entire Doppler distribution, such as extended cooling time with the chirp rate presented. The reduction of the number of cooled atoms indicated by our simulations (see Methods) will be offset by increasing the slow positron beam intensity.” to the concluding paragraph of the main text. (line 292 of the previous manuscript)
 - (65) We have added “(see Methods for the Doppler profile in the direction away from the surface of the aerogel; the current configuration did not provide cooling in this dimension)” to the sentence where we mentioned three-dimensional cooling. (line 292 of the previous manuscript)
 - (66) We have replaced “, which improves the systematic error due to the second-order Doppler shift” in the sentence describing the three-dimensional cooling with “, which reduces the second-order Doppler shift”. (line 293 of the previous manuscript)
 - (67) We have added “Despite the two-component velocity distribution, the light mass of Ps aids the spatial separation of the cooled component with time allowing for selective excitation.” to the concluding paragraph of the main text. (line 294 of the previous manuscript)
 - (68) We have removed “before laser cooling” from the sentence describing a precooling method for future experiments. (line 298 of the previous manuscript)
 - (69) We have replaced “It is necessary to increase the cooling rate by extending the cooling method to one that is not limited by spontaneous emission, as discussed for atomic and molecular cooling^{47,48}.” in the concluding paragraph with “Enhancing the cooling rate by adopting stimulated emission, as used in atomic and molecular cooling^{50,51}, could also be beneficial.” (line 299 of the previous manuscript)
 - (70) We have added “Doppler-broadened” to the sentence in the concluding paragraph describing the

importance of the future efficient three-dimensional cooling assisted by the stimulated emission.
(line 302 of the previous manuscript)

- (71) We have replaced “antimatter” with “particle-antiparticle” in the sentence describing the BEC of Ps in the concluding paragraph. (line 305 of the previous manuscript)
- (72) We have added the following mention to a related publication that was recently published at the end of the main text: “Note added: A recent publication⁵⁴ submitted on the same date reports one-dimensional cooling to 170(20) K using a broadband pulse (both results presented at the same international conference⁵⁵).”

Figures:

- (73) We have amended Fig. 1a to ensure that the frequencies between the sublevels of the 2P state are accurately represented.
- (74) We have revised Fig. 1b to clarify the setup within the vacuum chamber, including the magnetic lens.
- (75) We have made a modification to adopt Fig. 1c in the previous manuscript as an inset in Fig. 1b of the revised manuscript. Additionally, a timing chart has been introduced as Fig. 1c in the revised manuscript. Alongside these changes, the figure caption has been amended.
- (76) In the figure caption of Fig. 2a, we have added the following sentence: “ S_{off} is the 1S–2P excitation signal without cooling.” (line 173 of the previous manuscript)
- (77) In the figure caption of Fig. 2a, we have removed “We took into account the spectral width of the laser and the Lamb dip in this fitting function.” (line 179 of the previous manuscript)
- (78) We have modified the horizontal error bar displayed in Fig. 2b to represent both the experimental and fitting frequency resolutions.
- (79) We have replaced “Error bars originate from statistical and systematic errors in the measurement.” with “Error bars originate from statistical uncertainty in the number of ionised positrons.” in the figure caption of Fig. 2b. (line 177 in the previous manuscript)
- (80) We have replaced “in velocity distribution” with “in Doppler profile” in the figure caption of Fig. 2b. (line 182 in the previous manuscript)
- (81) We have added “(swept from -59 to -9 GHz)” to “... were set to a detuning suitable for cooling.”, and “(swept from -209 to -159 GHz)” to “... a control experiment with a larger detuning of the cooling laser” to the figure caption of Fig. 2b. (line 185 of the previous manuscript)
- (82) We have added (see Methods) to the sentence in which we introduced the dotted curve in the figure caption in Fig. 2b (line 187 of the previous manuscript).
- (83) We have added “In both the cooling and control experiments, spectrally narrowed probe pulses were employed.” to the figure caption of Fig. 2b. (line 189 of the previous manuscript)
- (84) We have replaced the title of Fig. 3 with “Numerical simulation.” (line 280 of the previous manuscript)
- (85) We have adopted Extended Data Fig. 4 in the previous manuscript as Fig. 3b. We added the following figure caption for Fig. 3b: “**b**, Reconstructed fractional change. The curve shows the simulated results. The thickness originates from the frequency resolution and the uncertainty in the uncooled fraction. The filled circles are from Fig. 2b.”
- (86) We have replaced “As the probability density of the 1S state at each velocity, we plotted relative to that of zero velocity at 0 ns.” in the figure caption of Fig. 3a with “The probability density at each velocity was plotted relative to that at 0 ns.” (line 284 of the previous manuscript)

Methods:

- (87) We have added the following sentence to the “Positronium generation” section of Methods: “The instability in the number of positrons remained at approximately 1% throughout the experiment.”

- (line 437 of the previous manuscript)
- (88) We have added a section titled “Cooling laser”, which briefly introduces the characteristics of the cooling laser, to Methods.
 - (89) We have edited the latter half of the first paragraph in the “Doppler spectroscopy” section in Methods. (line 471 of the previous manuscript)
 - (90) We have added “Because of the longitudinal multimode nature,” to the sentence “The spectral width of the second harmonic of the OPO was approximately 1.1×10^2 GHz.” in the second paragraph in the “Doppler spectroscopy” section in Methods. (line 485 of the previous manuscript)
 - (91) We have added the following sentence to the “Doppler spectroscopy” section in Methods: “Positron signals were observed in the range of approximately 30 ns to 80 ns after the pulse voltage application was initiated, corresponding to the drift time that is dependent on the Ps location at photoionisation.” (line 526 of the previous manuscript)
 - (92) We have added the following sentences to the “Doppler spectroscopy” section in Methods: “When the number of Ps in the whole velocity distribution was approximately 3×10^3 immediately after the production, the average number of detected positrons in the frequency-resolved measurements was typically 0.5. Consequently, the uncertainty in the excitation signal in the Doppler spectroscopy measurements was characterised by the randomness in the number of ionised positrons, which is governed by Poisson statistics.” (line 530 of the previous manuscript)
 - (93) We have added the following sentences to the “Doppler spectroscopy” section in Methods: “To achieve an adequate signal-to-noise ratio, it was necessary to set an appropriate measurement time. For the Doppler spectroscopy used in assessing the temperature of Ps gas with a frequency resolution of 110 GHz, the integration time for each probe frequency was approximately 20 minutes. During this integration time, the number of measurement cycles was approximately 1.2×10^4 . In the laser cooling experiment, where the resolution was set to be an order of magnitude greater and thus the signal was weaker, the integration time was approximately 4 hours for each probe frequency. During this time, the number of measurement cycles was approximately 7.2×10^4 .” (line 530 of the previous manuscript)
 - (94) We have replaced “A” with “C” in the definition of the model function for the measured Doppler profile in the “Analysis of the measured Doppler profile of uncooled Ps” section in Methods. (line 538 and 542 of the previous manuscript)
 - (95) We have added “(see the denominator in the fraction)” after the occupation probability of the excited state was defined in the “Analysis of the measured Doppler profile of uncooled Ps” section in Methods. (line 548 of the previous manuscript)
 - (96) We have replaced “Eq. (1)” with “ $S(\omega_R)$ ”, and “Eq. (2)” with “ P_e ” in the “Analysis of the measured Doppler profile of uncooled Ps” section in Methods. (line 550 of the previous manuscript)
 - (97) We have added a paragraph explaining the Doppler profile in the direction normal to the surface of the silica aerogel to the “Analysis of the measured Doppler profile of uncooled Ps” section in Methods. (line 563 of the previous manuscript)
 - (98) We have added “using a phenomenological model” to the title of the section “Analysis of the fractional change in the Doppler profile” in Methods. (line 564 of the previous manuscript)
 - (99) We have added the following sentences as the last paragraph of the “Analysis of the fractional change in the Doppler profile” in Methods: “We believe that the estimated population reductions were smaller than those expected from population reduction by laser cooling alone, due to the influence of delayed Ps release from the silica aerogel. Such delayed release from porous materials has been reported previously⁵⁷. Our empirical observations suggest the presence of Ps emitted from the silica aerogel several tens of nanoseconds after the injection of the positron bunch when we observed the components of zero lateral velocity. However, we cannot quantitatively discuss the delayed fraction due to the absence of systematic data available for presentation.” (line 601 of the previous manuscript)

manuscript)

- (100) We have replaced “The filled band shows the simulated result, whose interval at each relative frequency was determined by varying the frequency resolution from 8 GHz to 16 GHz.” in the “Numerical simulation” section in Methods with “The thickness of the curve was determined based on the frequency resolution within the range of 8 GHz to 16 GHz.” (line 700 of the previous manuscript)

Extended Data Figures:

- (101) We have added the Doppler profile in the direction normal to the surface of the silica aerogel as the Extended Data Fig. 1 and added the corresponding description to the figure caption.
- (102) The order of Extended Data Fig. 2 and Extended Data Fig. 3 in the previous manuscript has been switched, and their citations have been corrected accordingly.
- (103) We have enhanced the caption of Extended Data Fig. 2, now reordered as Extended Data Fig. 4, ensuring that there are no omissions in the definitions of the symbols within the figure.
- (104) We have replaced Extended Data Fig. 4, now reordered as Extended Data Fig. 5, with a figure that illustrates the simulated Doppler profile after cooling. The displayed frequency range has been expanded compared to the previous one. Alongside this change, we have added explanations to the “Numerical simulation” section of the Methods stating that components in the Doppler profile that do not resonate with the cooling laser are not affected.
- (105) We have added a set of figures as Extended Data Fig. 6 and the corresponding figure caption to illustrate dependence of the velocity distribution of the 1S state on the duration and the chirp rate of the cooling laser. We have added an explanation regarding this figure in the “Numerical simulation” section of Methods.

References:

- (106) We have added the following references:
D. B. Cassidy, Experimental progress in positronium laser physics, *Eur. Phys. J. D* **72**, 53 (2018).
A. Aspect *et al.*, “Laser cooling below the one-photon recoil by velocity-selective coherent population trapping.” *Phys. Rev. Lett.* **61**, 826 (1988).
H. Wallis and W. Ertmer, “Broadband laser cooling on narrow transitions”, *J. Opt. Soc. Am. B* **6**, 2211 (1989).
Glöggler, L. T. *et al.* Positronium Laser Cooling via the 1^3S-2^3P Transition with a Broadband Laser Pulse. *Phys. Rev. Lett.* **132**, 083402 (2024).
- (107) We have added the following mention to an international conference as Ref. 55. POSMOL 2023 (XXI International Workshop on Low-Energy Positron and Positronium Physics and XXIII International Symposium on Electron-Molecule Collisions and Swarms), presented on August 5, 2023.
- (108) We have replaced Ref. 41, which was a preprint with the published paper: Shu, K. *et al.* Development of a laser for chirp cooling of positronium to near the recoil limit using a chirped pulse-train generator. *Phys. Rev. A* **109**, 043520 (2024). The paper is now cited as Ref. 44.
- (109) We have added the following reference and cited in a paragraph in Methods where we described the delayed release from the silica aerogel: Cassidy, D. B., Hisakado, T. H., Meligne, V. E., Tom, H. W. K. & Mills Jr., A. P. Delayed emission of cold positronium from mesoporous materials. *Phys. Rev. A* **82**, 052511 (2010).

Acknowledgements:

(110) We have made minor modifications to Acknowledgements.

Other changes:

(111) Throughout the manuscript, we have made grammatical corrections and improvements to readability.

Reviewer Reports on the First Revision:

Referees' comments:

Referee #2 (Remarks to the Author):

This manuscript has been very much improved by the authors. They have addressed all of my detailed concerns and apparently also those of the other referee. In particular, the discussion is much easier to follow, and the figures are greatly improved. The authors are to be commended for this extensive and thoughtful revision of the first version of the manuscript. I can now recommend that the article be published in Nature, provided the editors have reviewed and accept the claim of simultaneous discovery with the AEGIS result. The technique presented here is innovative and shows great promise for the future.

Referee #3 (Remarks to the Author):

This is a brief report on a re-submission. The authors have adequately and comprehensively dealt with all the queries from the 2 referees who gave substantive comments on the first draft.

In particular, the doubts of referee 2 have been dealt with, and the extra information makes it clear that laser cooling of a significant fraction of the Ps population has been achieved.

This is a milestone achievement, well worth publishing in Nature.

I can also confirm that the basic result was reported at the POSMOL meeting in summer 2023. I attended the talk and my notebook attests to the result they reported.